# Visual projection neurons in the *Drosophila* lobula link feature detection to distinct behavioral programs

Ming Wu[†‡], Aljoscha Nern*[†], W Ryan Williamson, Mai M Morimoto, Michael B Reiser, Gwyneth M Card, Gerald M Rubin*

Janelia Research Campus, Howard Hughes Medical Institute, Ashburn, United States

**Abstract** Visual projection neurons (VPNs) provide an anatomical connection between early visual processing and higher brain regions. Here we characterize lobula columnar (LC) cells, a class of *Drosophila* VPNs that project to distinct central brain structures called optic glomeruli. We anatomically describe 22 different LC types and show that, for several types, optogenetic activation in freely moving flies evokes specific behaviors. The activation phenotypes of two LC types closely resemble natural avoidance behaviors triggered by a visual loom. In vivo two-photon calcium imaging reveals that these LC types respond to looming stimuli, while another type does not, but instead responds to the motion of a small object. Activation of LC neurons on only one side of the brain can result in attractive or aversive turning behaviors depending on the cell type. Our results indicate that LC neurons convey information on the presence and location of visual features relevant for specific behaviors.

*For correspondence: nerna@janelia.hhmi.org (AN); rubing@janelia.hhmi.org (GMR)

[†]These authors contributed equally to this work

Present address: [‡]Salubris Biotherapeutics, Inc., Gaithersburg, United States

**Competing interests:** The authors declare that no competing interests exist.

## Introduction

Many animals use vision to guide their interactions with the environment. Doing so requires their visual systems to extract information about the presence of ethologically relevant visual features from diverse and dynamic sensory landscapes. Most organisms with elaborated nervous systems compartmentalize this task; in vertebrates and insects, for example, visual processing begins in specialized brain regions of similar general structure, called, respectively, the retina and the optic lobe (*Sanes and Zipursky, 2010*). The signals computed in these early visual areas are then conveyed to different higher order brain regions by visual projection neurons (VPNs); ultimately these signals must be passed on to the neural circuits that control behaviors.

While VPNs are anatomically diverse and not necessarily closely related, their unique position as output channels of early visual centers makes these neurons attractive entry points for circuit-level analyses of visual processing. Studies of such neurons, for example of retinal ganglion cells in vertebrates and lobula plate tangential cells in insects, have provided insights into both the computations performed by the early visual system and the visual information that is available to higher brain regions (*Borst, 2014*; *Gollisch and Meister, 2010*). However, the relationship between signals encoded by the VPNs and visual behaviors has been difficult to systematically explore in any animal. Compared to photoreceptor neurons, which primarily respond to local luminance changes, VPNs can show much more specialized responses, some of which have been interpreted as encoding visual features directly relevant for specific behaviors, for example the presence of prey (*Lettvin et al., 1959*) or predators (*Zhang et al., 2012*). Here we present anatomical, behavioral and physiological analyses of Lobula Columnar (LC) neurons in *Drosophila* that support such a role for this class of VPNs.

**eLife digest** Many animals rely heavily on what they can see to interact with the world around them. But how does the brain use such visual information to guide behavior? Light-sensitive neurons in the eye cannot distinguish between the visual signals associated with, say, an approaching predator or a source of food. Yet the brain can make this distinction.

Networks of neurons in the brain perform computations to extract information from a visual scene that indicates the need for a particular behavior, such as an escape response. These networks are found in regions of the brain that communicate closely with the eyes. Cells known as visual projection neurons then relay the output of these networks to more central parts of the brain. By studying visual projection neurons, it is possible to work out what the eye tells the brain, and how the brain uses this information to control behavior. The fruit fly *Drosophila* is a suitable model organism in which to study these phenomena. This insect shows a range of behavioral responses to visual stimuli, and can be studied using sophisticated genetic tools.

Wu, Nern et al. set out to explore how a group of visual projection neurons known as lobula columnar cells help fruit flies respond appropriately to visual stimuli. Experiments revealed that individual subtypes of lobula columnar cells convey information about the presence and general location of specific visual features. Wu, Nern et al. identified a number of lobular columnar subtypes involved in triggering escape responses to specific stimuli – such as walking backwards or taking off in flight – as well as others that can trigger the flies to approach a target.

A next step is to map the circuits of neurons that act upstream and downstream of lobula columnar cells. This can help to reveal how these neurons detect specific visual features and how the fly then chooses and executes an appropriate behavior in response. Such studies in flies can provide insights into general principles of how brains use sensory information to guide behavior.

In flies, visual information is first processed in the optic lobes, which are comprised of four neuropils called the lamina, medulla, lobula and lobula plate (*Fischbach and Dittrich, 1989*; *Meinertzhagen and Hanson, 1993*). Each neuropil has a repetitive structure of several hundred retinotopically-arranged columns that support the parallel processing of visual signals from different points in space. Neurons projecting out of the optic lobes originate in the medulla, lobula and lobula plate with the majority from the latter two, deeper neuropil layers. The response properties of several lobula plate VPNs have been characterized in great detail, mainly through studies in larger flies (*Borst et al., 2010*; *Krapp et al., 1998*). These lobula plate tangential cells (LPTCs) show strongly directionally selective responses to a variety of motion stimuli and some LPTCs have been proposed to function as matched filters for the complex optic flow patterns associated with a fly's movements (*Krapp et al., 1998*). Many recent advances have also revealed key components of the upstream circuitry that provides LPTCs with their direction-selective response properties (reviewed in *Borst [2014]*). However, visual processing of stimuli other than wide-field motion is generally much less well understood. For example, flies respond to the movement, shape and position of objects (*Card and Dickinson, 2008b*; *Coen et al., 2016*; *Egelhaaf, 1985a*, *1985b*, *1985c*; *Ernst and Heisenberg, 1999*; *Götz, 1980*; *Liu et al., 2006*; *Maimon et al., 2008*; *Ofstad et al., 2011*; *Reichardt and Wenking, 1969*; *Robie et al., 2010*; *Tang et al., 2004*) or to different wavelengths of light (*Gao et al., 2008*; *Karuppudurai et al., 2014*; *Melnattur et al., 2014*). The neural substrates of these behaviors are largely unidentified but have often been proposed to involve neurons in the lobula. A major role of the lobula in visual processing is also indicated by anatomy, since the vast majority of medulla outputs project to the lobula (*Fischbach and Dittrich, 1989*).

Of the different classes of VPNs found in the lobula, one group, the LC neurons (*Fischbach and Dittrich, 1989*; *Otsuna and Ito, 2006*; *Strausfeld and Okamura, 2007*), has received particular attention. The LC neurons are the most numerous VPNs of the lobula and, as we show here, can be divided into over twenty distinct types. Anatomical characteristics of typical LC neurons are illustrated in *Figure 1*. Each LC type comprises multiple neurons of similar morphology whose individual dendritic arbors spread across only part of the array of lobula columns but which, with a few exceptions, cover the entire visual field as a population (*Fischbach and Dittrich, 1989*; *Otsuna and Ito,*

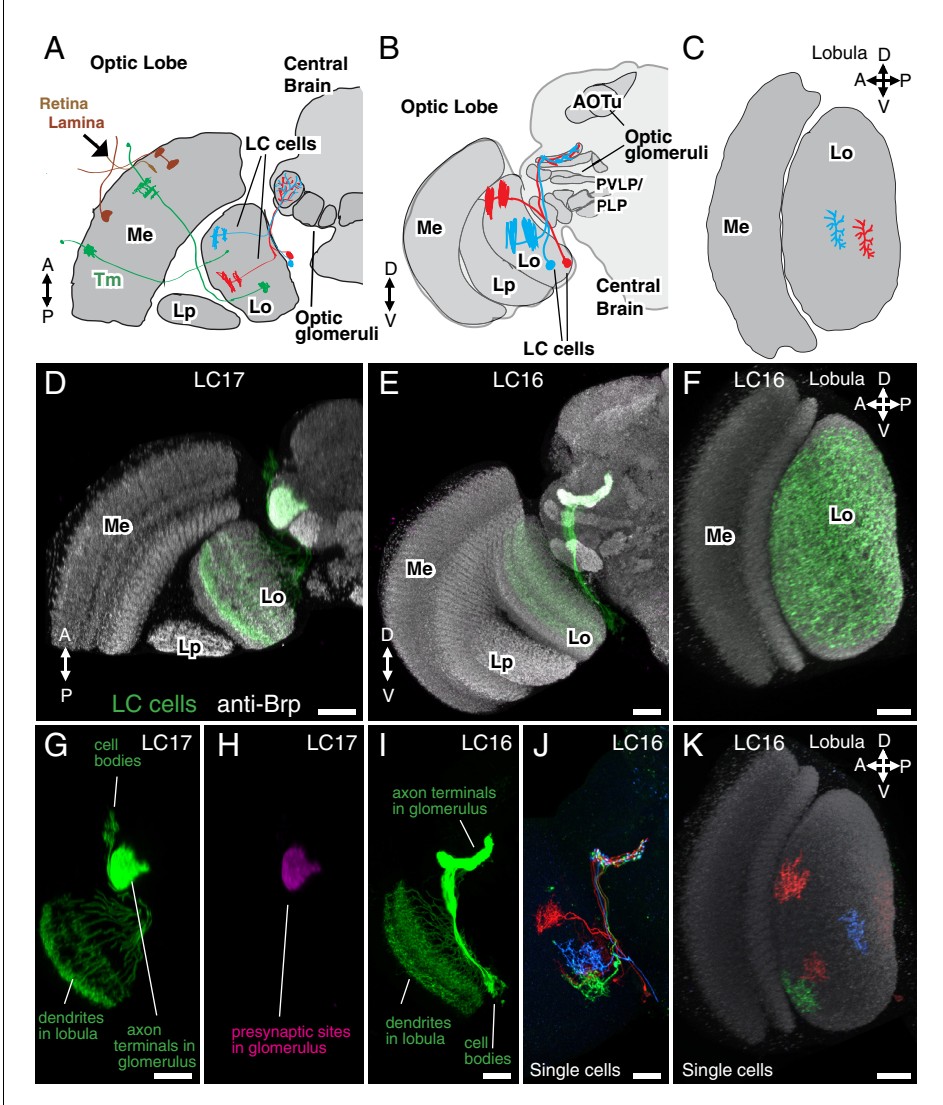

**Figure 1.** Introduction to lobula columnar (LC) neurons. Schematics (**A–C**) and confocal images (**D–K**) show the lobula and adjacent parts of the visual system. (**A,D,G,H**) Horizontal sections. (**B,E,I,J**) Anterior views. (**C,F,K**) Cross-section views of the lobula. Some subregions of the optic lobe (Me, Medulla; Lp, Lobula plate; Lo, Lobula) and central brain (AOTu, Anterior Optic Tubercle; PVLP, Posterior Ventrolateral Protocerebrum; PLP, Posterior Lateral Protocerebrum) are indicated in selected panels. Dendrites of individual LC neurons (red and blue cells in the schematics and red, blue and green cells in J,K) span only part of the visual field. As populations, the neurons of a given LC cell type cover most or all of the lobula (**F**), though LC neurons with regionally restricted lobula arbors also exist (e.g. LC14 (*Otsuna and Ito, 2006*), see *Figure 1—figure supplement 1*). LC neurons receive feed forward visual inputs from photoreceptors in the retina via a series of optic lobe interneurons (a few lamina neurons, in brown, and transmedullary neurons [Tm], in green, are illustrated as examples in [A]). This places LC neurons at least 2–3 synapses downstream of the photoreceptors. The majority of LC neurons projects to distinct target regions in the central brain called optic glomeruli; some of these are illustrated in (**A**) and (**B**) and also visible as distinct structures in the anti-Brp pattern in the images in (**D**) and (**E**). Most optic glomeruli are located in the PVLP and the adjacent more posterior PLP. The more dorsal AOTu (illustrated in [B]) is considered a specialized optic glomerulus. For a more detailed map of optic glomeruli see *Figure 3*. Confocal images show either populations of neurons (**D–I**) or individual cells labeled using Multicolor FlpOut (MCFO) (*Nern et al., 2015*) (**J,K**). LC cell types shown are LC17 (**D,G,H**) and LC16 (**E,F,I–K**). Population labeling (**D–I**) was with split-GAL4 driven expression of a membrane marker (green; myr::smFLAG, using pJFRC225-5XUAS-IVS-myr::smFLAG in *VK00005*) with a presynaptic marker also shown [magenta; synaptotagmin-HA, using pJFRC51-3XUAS-IVS-syt::smHA in *su(Hw)attP1*] in (**D,E**) and by itself in (**H**). A neuropil marker (anti-Brp) is included in grey in (**D–F,K**) and neuropil regions are also in grey in the schematics. Images in (**D,E,G–J**) were generated using brains that were

*Figure 1 continued on next page*

*Figure 1 continued*

computationally aligned to a template brain using the anti-Brp pattern as reference. The anti-Brp pattern in (D,E) is that of the standard brain used for alignment. Images in (D–K) show projection images of different views of three-dimensional image stacks; these were generated in either Fiji (http://fiji.sc/) (D,E,G–J) or Vaa3D (*Peng et al., 2010*) (F,K). Scale bars represent 20 μm.

The following figure supplement is available for figure 1:

**Figure supplement 1.** Examples of additional LC cell types and similar neurons.

---

*2006*; *Strausfeld and Okamura, 2007*) (*Figure 1C,F,K*). These anatomical properties contrast with those of LPTCs whose large dendritic arbors vary across lobula plate columns but are sufficiently stereotyped to be individually identifiable across animals. In addition to their dendritic arrangements, the second anatomical hallmark of most LC neurons is the convergence of their axons onto cell-type specific target regions in the central brain (*Otsuna and Ito, 2006*; *Strausfeld and Okamura, 2007*). These target regions represent distinct neuropil structures often referred to as 'optic glomeruli' (*Strausfeld, 1976*; *Strausfeld and Okamura, 2007*) (*Figure 1A,B,D,E,G–I*), named by analogy to the similarly shaped subunits of the antenna lobes, the olfactory glomeruli (*Laissue et al., 1999*). Similar to olfactory glomeruli, the optic glomeruli are enriched for synaptic sites and can be directly visualized with general neuropil markers (*Figure 1D,E*) (*Ito et al., 2014*). The most prominent of these optic glomeruli are found in the ventrolateral central brain, specifically the posterior ventrolateral protocerebrum (PVLP) and the posterior lateral protocerebrum (PLP) (*Figure 1B*). In addition, a small dorsal brain region with a glomerular structure, the anterior optic tubercle (AOTu) (*Figure 1B*), is considered a specialized optic glomerulus and also receives LC cell inputs (*Otsuna and Ito, 2006*; *Strausfeld and Okamura, 2007*).

LC neurons have been proposed to encode behaviorally relevant visual features (*Strausfeld and Okamura, 2007*), a conjecture that is largely based on the striking anatomical reorganization associated with the convergence of their axons into glomeruli. While the dendrites of different LC neuron types are arranged in overlapping retinotopic arrays in the lobula, their synaptic termini in the central brain are grouped into discrete glomeruli, most of which receive input from a single LC neuron type (*Otsuna and Ito, 2006*; *Strausfeld and Okamura, 2007*). The retinotopic structure of the lobula and other early visual neuropils is thought to facilitate the extraction of visual features from the spatiotemporal patterns of the activity of visual interneurons (ultimately going back to patterns of photoreceptor activity). By contrast, the organization of LC target regions into separate glomeruli, which is similar to the projection pattern of olfactory receptor neurons in the antennal lobe, suggests that different cell types already encode distinct features (such as the presence of specific odorants in the olfactory system) and that spatial information is of secondary importance.

Though recent studies have begun to explore functional properties of LC neurons (*Aptekar et al., 2015*; *Mu et al., 2012*), the hypothesis that LC neurons are feature responsive cells remains largely untested and little is known about LC neuron function in general. One limitation has been experimental access to defined LC types: although several types of LC neurons have been described in *Drosophila* (*Fischbach and Dittrich, 1989*; *Otsuna and Ito, 2006*), specific genetic reagents for the study of these neurons were largely lacking and inputs to several prominent optic glomeruli had not been identified, though candidates for new LC neuron types are, for example, recognizable in images of fly brain clonal units (*Ito et al., 2013*).

In parallel to our work, two recently published independent studies (*Costa et al., 2016*; *Panser et al., 2016*) have made use of existing image data (such as (*Chiang et al., 2011*; *Jenett et al., 2012*)) to reveal additional VPN pathways from the lobula to optic glomeruli. Costa et al. report the identification of several new LC neuron types as one of several examples of the application of a new computational method that groups similar neurons using aligned brain images. Their work illustrates the potential power of these computational tools but places less emphasis on the description or interpretation of specific findings regarding VPN neuroanatomy. Panser et al. use a different computational method to identify GAL4 driver lines from the Janelia and Vienna Tiles collections (*Jenett et al., 2012*; *Kvon et al., 2014*) that appear to include expression in VPN inputs to optic glomeruli and use these lines to generate an anatomical map of these glomeruli.

In this study, we apply a previously established genetic intersectional approach (*Aso et al., 2014a*; *Tuthill et al., 2013*) to generate highly specific split-GAL4 (*Luan et al., 2006*; *Pfeiffer et al., 2010*) driver lines that target different LC types. We first use these lines to provide detailed anatomical descriptions of 22 different LC types about half of which had not been previously described by Otsuna and Ito (*Otsuna and Ito, 2006*). For each type, we examine not only the position and shape of the target glomerulus but also many other anatomical features (such as lateral spread and layer patterns of lobula arbors, cell body positions and cell numbers), significantly extending the analyses in other studies. We find that each LC type is characterized by a distinct layer distribution of dendrites in the lobula and, with a few exceptions, a unique axonal output region in the central brain. Comparison of these output regions with the pattern of optic glomeruli indicates that most prominent glomeruli are the target regions of a specific LC type. Our glomerulus map largely concurs with the results of Panser et al. but is based on higher resolution images that allow us to better separate adjacent glomeruli and to define the target regions of two additional LC cell types. We also show that another LC type can be subdivided into four anatomically and genetically defined subtypes. Independent of the overall anatomical transformation associated with the convergence of LC neuron axons into glomeruli (see above), LC neuron axons of a given type might either retain or discard retinotopy within their target glomerulus. Potential retinotopy within optic glomeruli has not been examined in detail and images with sparse labeling of LC neurons have been interpreted as arguing either for or against such axonal retinotopy (*Otsuna and Ito, 2006*; *Panser et al., 2016*). To further explore possible retinotopy within individual glomeruli, we used multicolor stochastic labeling of individual LC neurons. In general, we did not observe detectable retinotopy of LC neuron axons within a glomerulus. However, we did identify a few exceptions; in particular, axonal projections of LC neurons to the AOTu retain retinotopy for azimuthal positions, suggesting a specialized role of the AOTu in the processing of spatial information. Stochastic labeling also allowed us to examine additional anatomical features of LC cells such as the dendritic arbor size and shape that cannot be observed at the population level. We also used our new driver lines to explore behaviors associated with LC neuron activity, by examining the response of freely behaving flies to optogenetic depolarization of individual LC types. In several cases such activation triggers distinct, highly penetrant behavioral responses that resemble natural, visually guided behaviors. In particular, using high-speed videography we show that two of these evoked behaviors, flight-initiating jumping (takeoff) and backward walking, resemble natural avoidance behaviors that can be elicited by a looming visual stimulus. Moreover, the two LC types whose activation evokes these avoidance behaviors respond to looming stimuli as assayed by two-photon calcium imaging. The encoding of looming is not a feature of all LC types, as we found a third LC type to be selectively responsive to small object motion. Finally, we present evidence that activation of LC neurons on only one side of the brain can induce attractive or aversive turning behaviors, depending on the cell type. Taken together, our anatomical and functional data suggest that each LC type conveys information about the presence and at least general location of a behaviorally relevant visual feature. Although details of our data suggest further downstream integration of signals from different LC types, the highly penetrant phenotypes we observe with activation of some LC types are consistent with a simple model for the initiation of several behaviors.

## Results

### Characterization of visual projection neurons that connect the lobula with glomerular target regions in the ipsilateral central brain

To study and further identify LC neurons, we screened collections of GAL4 driver lines (*Jenett et al., 2012*; *Kvon et al., 2014*) (Barry J. Dickson, personal communication). We searched for cell types that consisted of many similar cells which as populations covered the entire array of visual columns in the lobula and whose axonal projections converged onto single glomerulus-like regions in the ipsilateral central brain (*Figure 1*). Other types of lobula VPNs were also identified in the screen but will not be further characterized here. These included a number of additional LC-like cell types which were excluded here because of the different structure or location of their target regions or because their combined dendrites appeared to be restricted to lobula subregions corresponding to only part of the visual field. Some examples of such cells, which include the previously described LC14

(*Hassan et al., 2000*; *Otsuna and Ito, 2006*), are shown in *Figure 1—figure supplement 1*. In addition to neurons having dendrites in the lobula, we identified columnar VPNs associated with other optic neuropils that also had glomerular target regions (see *Figure 1—figure supplement 1* for an example) but we excluded them from further analysis. For the cell types that met our criteria, we used the split-GAL4 intersectional approach (*Luan et al., 2006*; *Pfeiffer et al., 2010*), where GAL4 activity is restricted to the overlap in the expression patterns of two GAL4 driver lines, to generate driver lines with predominant or exclusive expression in individual LC types. In combination, the split-GAL4 driver lines reported here have expression in 22 different types of LC neurons. Seven of these LC types (LC4, LC6, LC9-LC13) have been previously described (*Fischbach and Dittrich, 1989*; *Otsuna and Ito, 2006*). For consistency, we named new LC types by extending a previously used numbering scheme (*Otsuna and Ito, 2006*) and coordinated these names with another group that also found, and very recently reported (*Panser et al., 2016*), several of the new LC neurons described here; except for the cell types shown in *Figure 1—figure supplement 1* (LC14, LC19, LC23), the gaps in the sequence of LC cell type names (LC1-LC3, LC5, LC7 and LC8) are due to the naming scheme and do not correspond to known LC types not covered in this study. Anatomical characteristics of the different LC neuron types are described below; for genotypes of the split-GAL4 driver lines see Materials and methods. Overall expression patterns of the main split-GAL4 lines used in this study can be found in *Figure 2* and *Figure 2—figure supplement 1*. Expression patterns of some additional lines for these same cell types are shown in *Figure 9—figure supplement 1* and *Figure 10—figure supplement 1*. Confocal stacks of all lines can be downloaded from www.janelia.org/split-GAL4. Details of which split-GAL4 driver lines and other transgenes were used in individual experiments are provided in the Materials and methods and in *Supplementary file 1B, D*.

We first present detailed anatomical studies of the LC cell types labeled by our split-GAL4 driver lines (*Figures 1–7*). We then focus on LC neuron function (*Figures 8–13*). To anatomically characterize LC neuron types and to confirm the identity of the LC neurons labeled by each split-GAL4 driver line, we examined cell shape (*Figures 2* and *3*; examples shown in *Figure 1D,E,G,I*) and the location and shape of target regions in the central brain (*Figure 3*; visualized with a presynaptic marker [HA-tagged synaptotagmin; syt-smHA]; *Figure 1H* shows an example) for each LC cell population. We also carried out stochastic labeling experiments to reveal the morphology of individual cells (illustrated in *Figure 1J,K*) and to explore the arrangements of LC neuron axon terminals relative to the retinotopic positions of their dendrites (*Figure 4*). Several stereotyped morphological features revealed in these experiments support the classification of LC neurons into the cell types described here: the shape and location of the target glomerulus as well as the size, shape and layer pattern of the lobula arbors; the approximate position of cell bodies; and the path followed by the axons. The anatomical features of LC neurons are summarized in a table in *Supplementary file 1A*. This table also includes estimates of the approximate number of cells per LC type for most LC types. We describe in detail the distribution and structure of axonal target regions of LC neurons in the central brain (*Figure 3*) and the layer pattern, size and shape of their dendrites (*Figures 5*, *6* and *7*) as these features provide information about the potential synaptic partners of each LC type.

## LC neurons are the main visual inputs to optic glomeruli

We used split-GAL4 driven expression of syt-smHA to visualize presynaptic sites of individual LC neuron populations in the central brain (*Figure 1H*). To facilitate visualization of combinations of LC cell types, we aligned data collected from confocal stacks of individual driver lines to a standard brain (*Aso et al., 2014a*). Together, the LC neurons characterized in this study project to 19 distinct target regions in the ipsilateral central brain (*Figure 3*, *Videos 1* and *2*). In addition to the shape and location of these target regions, the axonal paths followed by LC neurons are also stereotyped with individual cells of the same type showing a similar projection pattern (*Figure 2* and *Figure 3A, D,G*). To compare LC neuron target areas to the position of optic glomeruli, we focused on the PVLP where several large glomeruli can be readily visualized with general markers of synaptic density (*Ito et al., 2014*). LC neuron target regions in the PVLP visualized by expression of a presynaptic marker in LC neurons closely matched the glomerular neuropil pattern revealed by anti-Brp staining, which labels presynaptic active zones (*Wagh et al., 2006*) (*Figure 3B,C* and *Figure 3—figure supplement 2*, *Video 1*). This result confirmed optic glomeruli as the target regions of these LC neurons and identified LC input neurons for each of several glomeruli whose projection neuron inputs were

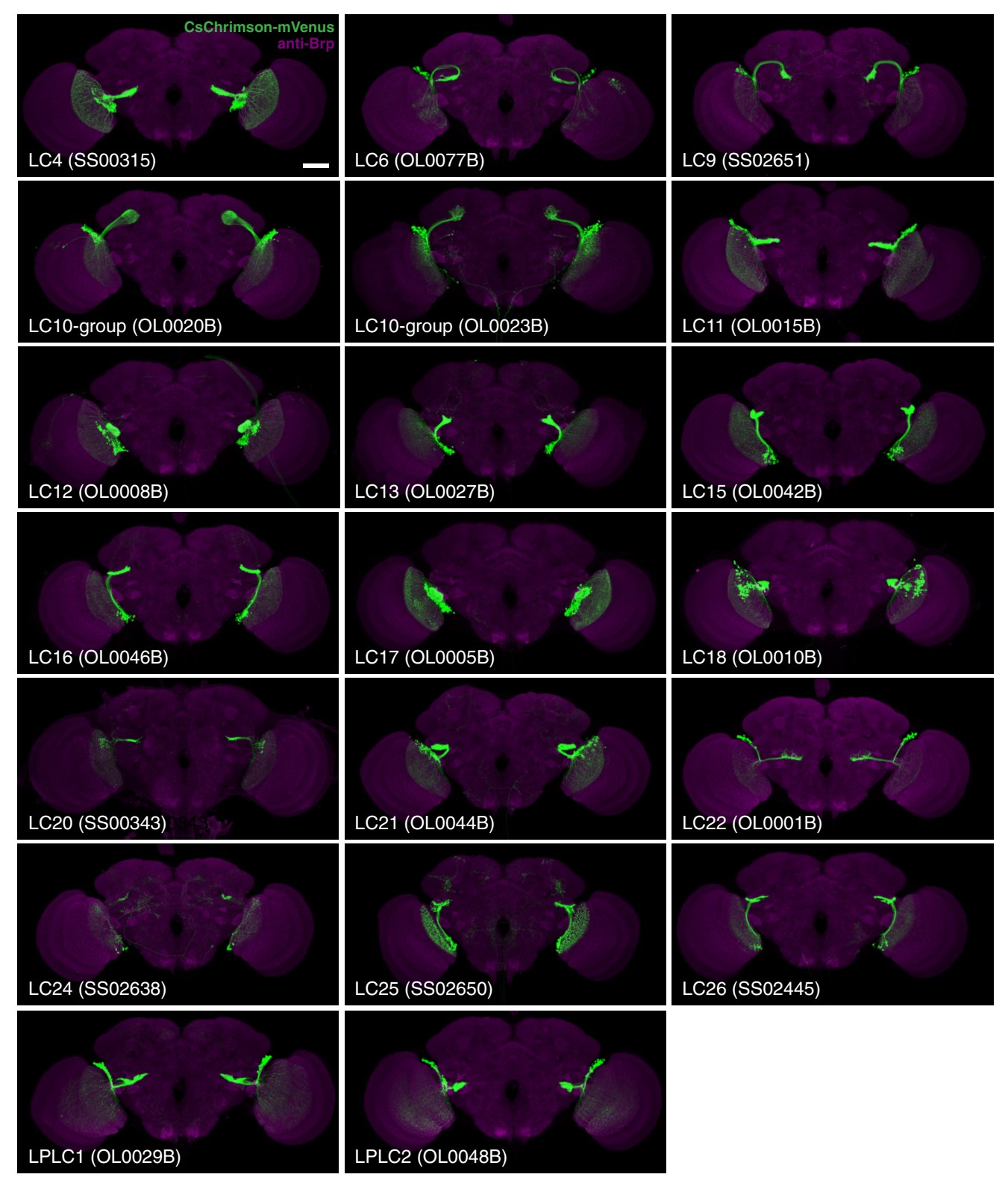

**Figure 2.** Expression patterns of LC neuron split-GAL4 lines. Split-GAL4 driven expression of 20xUAS-CsChrimson-mVenus (insertion in *attP18*; visualized using anti-GFP antibody labeling; green) and a neuropil marker (anti-Brp, magenta) are shown. Genotypes are identical to those used in the behavioral experiments in *Figure 8*. Some adjustments of brightness and contrast of individual samples were made. For each driver line (but not across different lines), adjustments and microscope settings were identical for the brain shown in this Figure and the corresponding ventral nerve cord (VNC)

*Figure 2 continued on next page*

*Figure 2 continued*

shown in *Figure 2—figure supplement 1*. The images in this and other figures with CsChrimson expression patterns are representative of 2–5 brains and 2–3 VNCs imaged for each split-GAL4 line. Scale bar represents 50 µm. Original confocal stacks are available from www.janelia.org/split-GAL4.

The following figure supplement is available for figure 2:

**Figure supplement 1.** VNC expression patterns of the corresponding brains shown in *Figure 2*.

unknown. In particular, our data show that each of the 12 most prominent glomeruli in the PVLP (or, in one case, the boundary region of PVLP and PLP) (*Figure 3B,C*, *Video 1*) is the target of one columnar VPN type from the lobula. Ten of these glomeruli are the targets of typical LC cell types while two glomeruli receive inputs of neurons with dendrites in both lobula and lobula plate; we refer to these LC-like cells as LPLC1 and LPLC2. Similar cells have been described in other Diptera (*Douglass and Strausfeld, 1998*). Although we visually screened the expression patterns of several thousand GAL4 lines, we did not observe additional cases where the terminal fields of a VPN similarly coincided with one of these glomeruli (as defined by anti-Brp labeling), suggesting that LC neurons are the primary inputs to these structures.

The target regions of most LC cell types in the PVLP were clearly distinct with no or only minimal overlap (*Figure 3B,C*); some glomeruli appear to overlap in the projection image in *Figure 3B* but occupy distinct three-dimensional positions (see *Videos 1* and *2*). In addition to projections to the large PVLP glomeruli, we also characterized several LC neuron types that converged onto more posterior, often smaller regions in the PVLP and PLP (*Figure 3D–F*, *Videos 1* and *2*). The target regions of these additional LC cell types appeared more variable in shape and arrangement. Finally, several LC neuron types project to the large subunit of the AOTu (*Figure 3G–I*); the properties of these LC10 cells are discussed further below. In sum, our results provide a high resolution map of LC neuron target regions and establish a link between the pattern of glomeruli detected with general neuronal markers and the neuron projections of specific LC cell types. With a few exceptions, primarily the AOTu (discussed further below) and the target regions of LC25 and LC22, each glomerulus described here can be uniquely identified as the target region of a single LC neuron type that appears to be the major input to that glomerulus. Although other, non-columnar VPNs may also have some presynaptic sites in or near these optic glomeruli, we did not observe similarly prominent non-LC input neurons to these structures in our GAL4 line screen.

## Most optic glomeruli do not show obvious internal retinotopy of their LC neuron inputs

A prominent feature of LC neurons is the change in the anatomical arrangement of LC neurons as they project from the lobula to the central brain. In the lobula, LC neuron dendrites form parallel retinotopic arrays, whereas their synaptic termini in the central brain are grouped into discrete glomeruli. However, it has not been examined in detail whether any retinotopy is preserved in the glomeruli; that is, whether the positions of the dendrites of individual LC neurons in the lobula and of their presynaptic terminals within their target glomerulus are correlated. To directly compare the relative positions of the dendrites of different LC cells of the same type in the lobula with the arrangement of the presynaptic arbors of the same cells, we examined samples in which several LC cells of the same type were stochastically labeled in distinct colors (*Figure 4*) using the Multicolor FlpOut (MCFO) technique (*Nern et al., 2015*). These experiments revealed that most individual LC neurons, with the exceptions discussed below, had branched terminals that appeared to spread throughout their target glomerulus without obvious spatial restriction to subregions or correlation between the distributions of their dendritic arbors in the lobula and their presynaptic arbors in the glomerulus. *Figure 4A–C* shows examples of this analysis for LC16. Qualitatively, most other LC cell types appeared similar to LC16 in that they lacked any obvious preservation of retinotopy at the glomerulus level; each individual cell's axonal terminal was intermingled with others, featuring pre-synaptic boutons throughout the glomerulus (see *Figure 4—figure supplement 1* for examples of additional LC cell types). Of course, retinotopic patterns in the synaptic connections between these LC neurons and their targets that are not apparent at the resolution examined here may exist. LC9 is

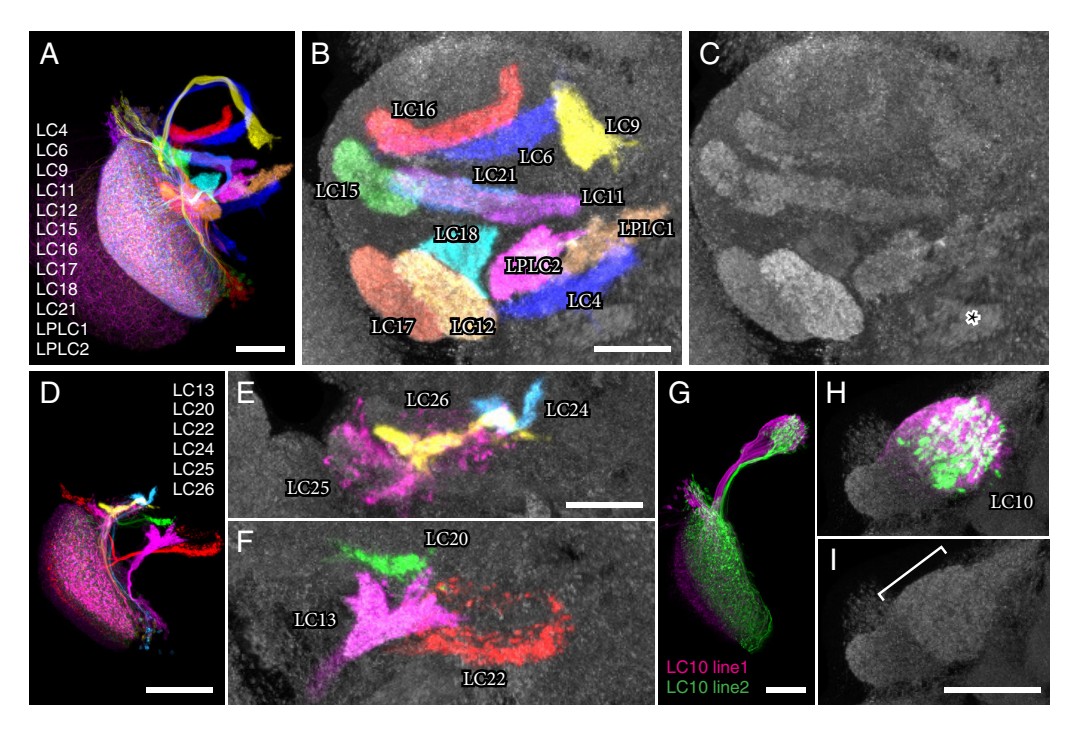

**Figure 3.** LC neuron terminals in the central brain are organized into distinct neuropil structures. (A) Illustration of the projection patterns of 12 LC cell types that project to major optic glomeruli in the PVLP (or in the PVLP/PLP boundary region). Image is a substack maximum intensity projection of a composite image stack generated from 12 computationally aligned image stacks (one for each cell type). Images were manually segmented to exclude background and some off-target cell types. Unedited pre-alignment stacks are available from www.janelia.org/split-GAL4. For details of genotypes see *Supplementary file 1D*. (B,C) Target regions of the LC neurons shown in (A) match the optic glomeruli pattern in the PVLP. Target regions of different LC cell types were labeled by split-GAL4 driven expression of a presynaptic marker (pJFRC51-3XUAS-IVS-syt::smHA in *su(Hw)attP1*, detected using anti-HA antibody labeling). Images for different cell types were edited and combined as described above. The anti-Brp pattern of the standard brain used for alignment is shown in grey. (C) Pattern of optic glomeruli revealed by anti-Brp labeling. Image is the same as the anti-Brp channel of the overlay shown in (B). Note the close correspondence of the presynaptic terminals of LC cell populations and optic glomeruli. Asterisk marks a large synapse rich (based on anti-Brp labeling) glomerular structure in the PLP that appears to be the target of several columnar VPNs that were not included here since we considered them to be primarily associated with the lobula plate, not the lobula (though some have lobula branches). As an example, one cell of one such type (LPC1) is shown in *Figure 1—figure supplement 1E*. LPC1 was also identified by (*Panser et al., 2016*). (D–F) Overlays generated as in (A) and (B) showing the projection patterns (D) and target regions (E,F) of additional LC neurons with terminals in the posterior PVLP and in the PLP. LC24, LC25 and LC26 projected to locations close to those of LC15, LC6 and LC16 but slightly more posterior and might also slightly overlap with each other. In particular, LC25 was unusual in that its terminals spread along the surface of, and perhaps partly overlapped with, the LC15 target region and other adjacent glomeruli (E and *Figure 3—figure supplement 2*). Similarly, part of the boundary of the LC22 target region, as visualized by synaptic marker expression in LC22 cells, was less well defined than the boundaries of most other glomeruli (F). The LC22 glomerulus also appears to overlap with the target region of a second columnar VPN: stochastic labeling experiments revealed an additional LPLC-like cell type (distinct from LPLC1 and LPLC2 and tentatively named LPLC4 in agreement with [*Panser et al., 2016*]) that projects to the same approximate location as LC22. While we have yet to generate specific split-GAL4 drivers for this additional LPLC cell type and therefore did not further characterize these neurons here, their overlap with LC22 terminals was directly confirmed by co-labeling of single cells of both types in the same specimen (*Figure 3—figure supplement 1*). (G–I) LC10 neurons project to the large subunit of the AOTu. Overlays (generated as described above) showing cell shapes (G) and presynaptic sites (H) of LC10 cells labeled by two different split-GAL4 driver lines. LC10 neurons showed two distinct general projection paths with axons entering the large subunit of the AOTu from both dorsally and ventrally (magenta cells) or only from ventrally (green cells), in agreement with previous findings suggesting the existence of LC10 subtypes (*Otsuna and Ito, 2006*). (I) Anti-Brp reference pattern alone. The LC10 terminals are in a distinct large subcompartment (white bracket) of the AOTu. The AOTu also includes smaller subunits adjacent to this LC10 target region. Scale bars represent 30 μm (A,G,I), 20 μm (B, E) or 50 μm (D).

The following figure supplements are available for figure 3:

**Figure supplement 1.** A second type of columnar VPN projects to the LC22 target region.

**Figure supplement 2.** Presynaptic marker expression in individual LC cell types.

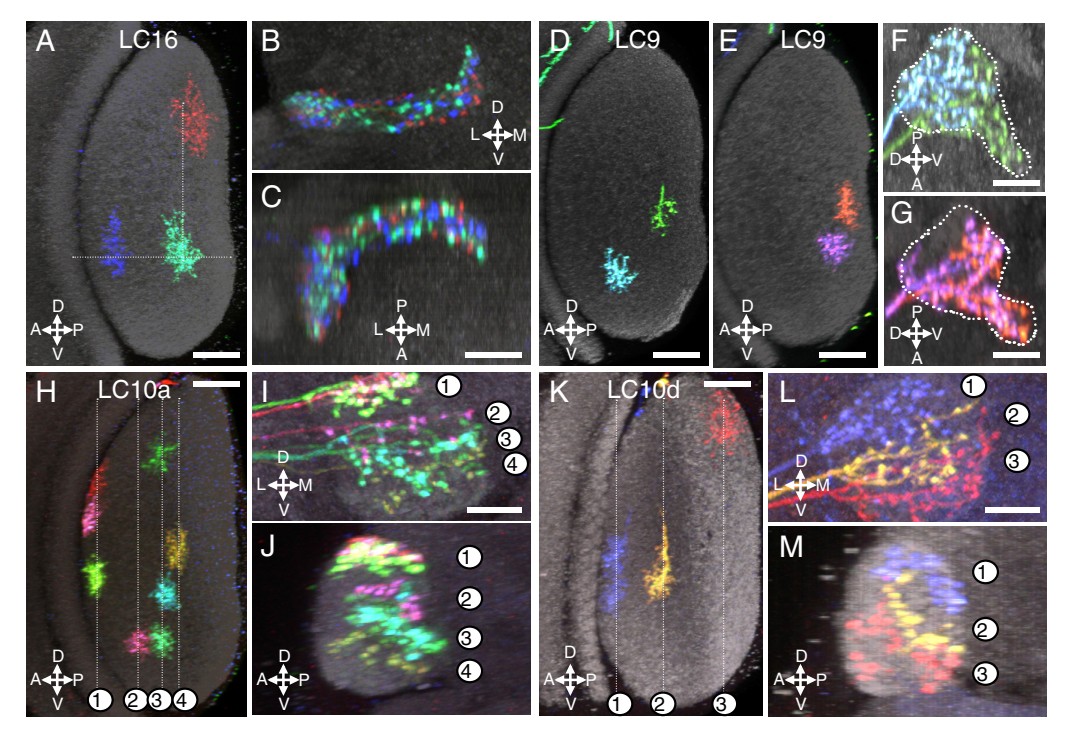

**Figure 4.** Multicolor stochastic labeling reveals differences in the arrangement of the terminal arbors of different LC types within their target glomerulus. (A–C) LC16. (A) Position of the dendrites of three LC16 cells in the lobula in a layer cross-section view. Each cell occupies a distinct position along the long (DV) and short (AP) axes of the lobula (dotted white lines were added to facilitate comparison of the positions of different cells). By contrast, within the target glomerulus (B,C; two roughly orthogonal views are shown) the arbors of the same cells are intermingled without an obvious correlation to the retinotopic pattern in the lobula. Most other LC types appeared similar to LC16; additional examples are shown in *Figure 4—figure supplement 1*. (D–G) LC9. Most LC9 cells have presynaptic arbors that spread through more than one half of the glomerulus but do not fill it completely (F,G). Comparison of the approximate positions of arbors of individual LC9 cells in the glomerulus (F,G) with the AP positions of their dendrites in the lobula (D,E) suggests some retinotopy within the glomerulus, though with very low spatial resolution: For example, out of 57 examined LC9 cells, all of the cells (27/27) with axon terminals in the anterior-ventral tip of the glomerulus (such as the green cell in [F] and both cells in [G]) also had dendrites in the posterior half of the lobula while this was the case for only a few of the remaining LC9 cells (6/30). White dotted lines in (F) and (G) indicate the approximate boundaries of the LC9 glomerulus. (H–M) LC10. The relative order of LC10a (H–J) and LC10d (K–M) terminals along the DV axis of the AOTu (shown in two orthogonal views, I,J for LC10a and L,M for LC10d) matches the order along the AP axis of the lobula (H for LC10a and K for LC10d). Individual cells were labeled using MCFO. Reconstructed views of reoriented substacks generated in Vaa3D are shown. For both LC10a and LC10d, similar results were observed for LC10 cells from five optic lobes, each with at least three labeled cells. Analyses of MCFO-labeled LC10b (36 cells from 18 brains) and LC10c (33 cells from 17 brains) single cells also showed an approximate correspondence between AP positions of dendrites in the lobula and DV positions of axonal arbors in the AOTu. LC10b cells also showed considerable variation in their lateral-medial spread within the medial zone of the AOTu but further analyses will be required to explore possible correlations between these differences and arbor positions in the lobula. Scale bars represent 20 µm (A,D,E,H,K) or 10 µm (C,F,G,I,L).

The following figure supplement is available for figure 4:

**Figure supplement 1.** Terminal arbor arrangements of additional LC cell types.

an example of a cell type that appeared to retain some retinotopy, though with very low spatial resolution, at the level of the axonal terminals: terminals of single LC9 cells expanded through only part of the glomerulus and their position correlated with the approximate position of the corresponding dendrites in the lobula (*Figure 4D–G*).

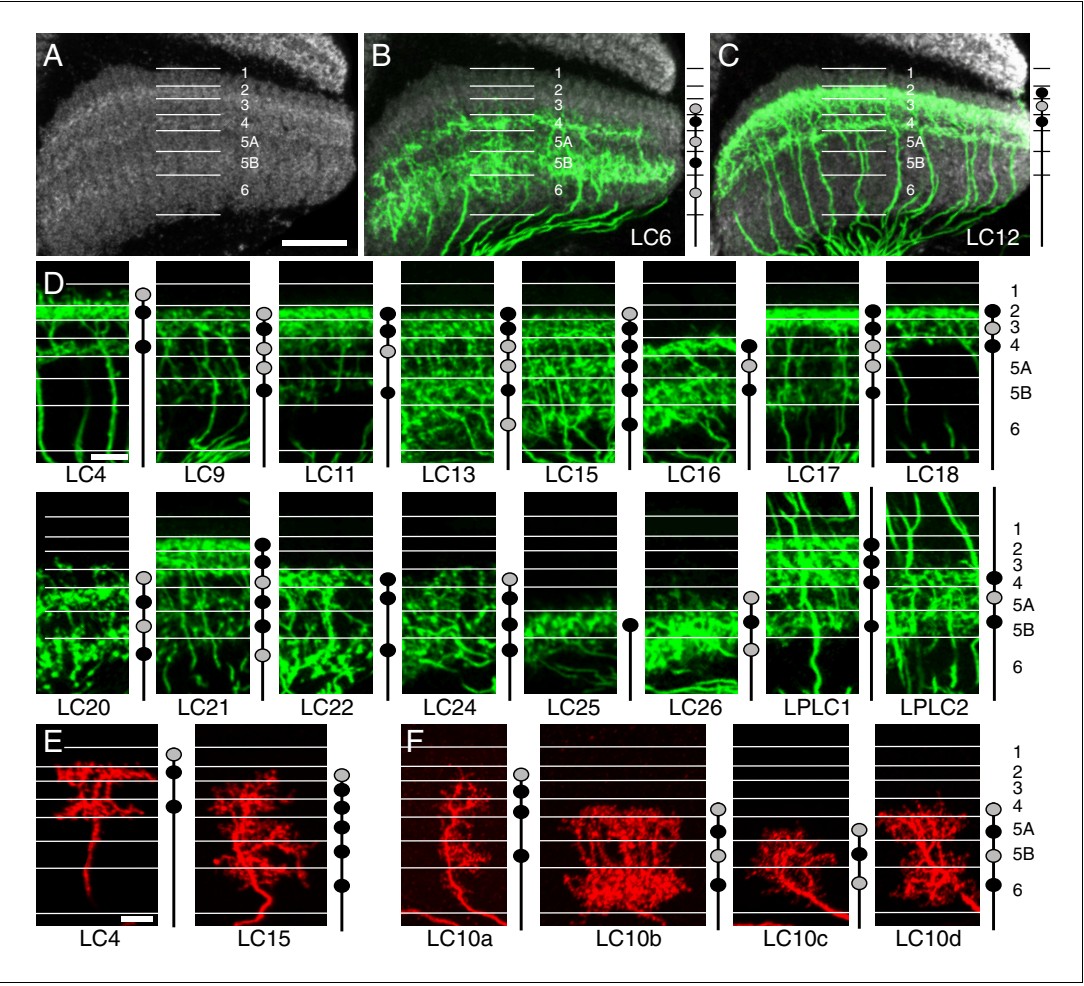

**Figure 5.** Layer specific arborizations of LC neurons in the lobula. (**A**) Anti-Brp neuropil marker shows bands of different intensity in the lobula that can serve as approximate markers of layer boundaries. The image is a maximum intensity projection through 10 adjacent sections (0.38 μm spacing) of the reference channel of the standard brain used for alignments. Approximate layer boundaries are indicated. Layer boundaries were defined by the positions of known cell types and closely match the anti-Brp pattern (see *Figure 5—figure supplement 1*). Lo5 was divided into two sublayers based on the anti-Brp pattern. Further subdivisions of strata based on the positions of arbors of different cell types would be possible but were not applied here. (**B,C**) Layer patterns of LC6 (**B**) and LC12 (**C**). Split-GAL4 expression of a membrane marker is shown in green. Images were aligned to a standard brain using the anti-Brp pattern (shown in grey). Images are maximum intensity projections through the same series of sections of brains aligned to the same template as in (**A**). Approximate layer boundaries are marked with white lines. (**D**) Layer patterns of the remaining LC cell types (except LC10 neurons). Projections were generated as in (**B,C**) but are shown without the anti-Brp pattern. All layers, but only a portion of the lobula is shown. Schematics in (**B–F**) indicate innervated layers as filled circles; black circles represent more extensive arborizations than grey circles. Note that these simplified schematics do not capture some details of the layer patterns (such as sublayer patterns). An additional description of layer patterns can be found in *Supplementary file 1A*. (**E**) Single cell layer patterns are consistent with layer patterns seen at the population level. LC4 and LC15 are shown as examples. Additional single cell images can be found in *Figure 7—figure supplement 1*. (**F**) Layer patterns of LC10 subtypes. LC10b and LC10d cells have similar layer patterns but differ in other aspects such as arbor size (LC10b arbors in the lobula are larger). Additional examples of MCFO labeled LC10 cells of different subtypes can be found in *Figure 7—figure supplement 1* and *Figure 10—figure supplement 2*. Images in (**E,F**) were manually matched to the layer markers using the anti-Brp pattern. The scale of some images was slightly adjusted to compensate for varying depth of the lobula. Scale bars represent 20 μm (**A**, also applies to **B** and **C**) or 10 μm (**D,E**; scale bar in **E** also applies to **F**).

The following figure supplements are available for figure 5:

*Figure 5 continued on next page*

*Figure 5 continued*
**Figure supplement 1.** Layer positions of arbors of known cell types.
**Figure supplement 2.** Potential presynaptic sites of LC neurons in the lobula.

## The AOTu is innervated by multiple LC cell types that preserve some retinotopy

The most striking example of retinotopy within a glomerulus was the arrangement of the terminals of the LC10 neurons that project to the AOTu (*Figure 4H–M*). LC10 cells have branched terminals similar to other LC neurons but with limited dorso-ventral (DV) spread so that individual cells did not cover the entire AOTu along this axis (*Figure 4I,J,L,M*). In the AOTu, the relative positions of presynaptic arborizations of LC10 cells along the DV axis of the AOTu largely matched the order of their dendrites along the anterior-posterior (AP) axis of the lobula (*Figure 4H,K*), though with more overlap between arbors in the AOTu compared to the lobula. By contrast, the DV positions of LC10 dendrites in the lobula were not obviously correlated with arrangements of neuronal processes in the AOTu and the positions of different LC10 terminals largely overlapped along the lateral-medial (LM) and AP axes of the tubercle (*Figure 4I,J,L,M*). LC10 cells also differed in a second key respect from LC cell types innervating most other optic glomeruli; consistent with previous reports (*Otsuna and Ito, 2006*), we found that the AOTu is the target of axonal projections of several anatomically distinct LC10 subtypes. Our classification of LC10 cells is mainly based on the size and layer patterns of the lobula arbors of these neurons; these differences, which are predicted to reflect different presynaptic inputs to these cells, are described in detail below (*Figure 5F*, *Figure 7—figure supplement 1* and *Figure 10—figure supplement 2*). Previously reported subtype distinctions (*Otsuna and Ito, 2006*) were based on differences of the axonal path (dorsal or ventral) within the AOTu, which we also observed but which do not unambiguously identify the subtypes described here, and distinct lateral-medial positions of LC10 terminals, which we did not find evidence for; all LC10 types labeled

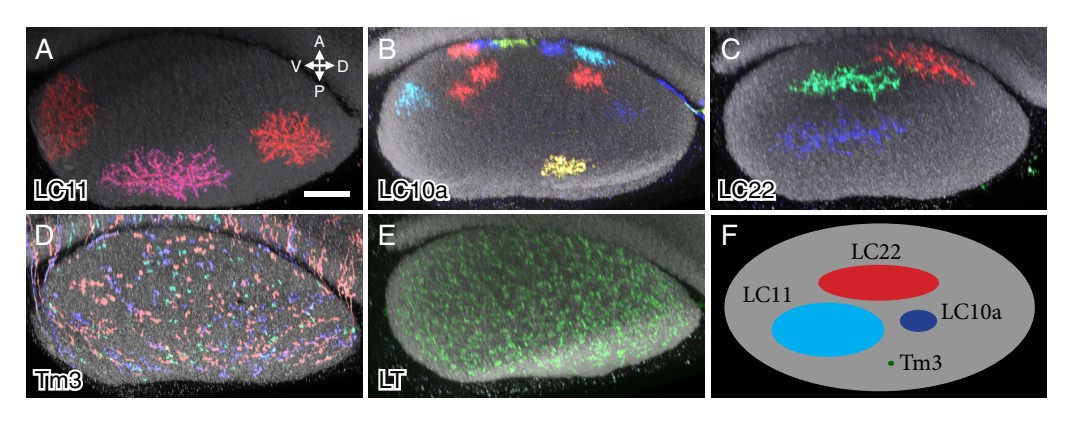

**Figure 6.** Column spread of LC neurons and other cell types in the lobula in cross-section views. Cells were labeled using MCFO. Cross-section views of the lobula were generated using Vaa3D. The AP and DV axes of the lobula are indicated. Anti-Brp reference marker is shown in grey. LC neuron arbor sizes and shapes in the lobula are diverse across different cell types but similar within each type. LC cell types shown are LC11 (**A**), LC10a (**B**) and LC22 (**C**). The remaining LC cell types are shown in *Figure 6—figure supplement 1*. MCFO labeled cells of a columnar medulla neuron type (Tm3) present in each of ~750 visual columns (**D**) and a lobula tangential cell (LT) that spans the entire lobula (**E**) are shown for comparison. All LC arbors are multicolumnar with estimated sizes from about 10 (LC10a) to over 60 (LC11) visual columns. LC22 cells were similar in size to LC11 along the long (DV) but not the short (AP) axis of the lobula. (**F**) Schematic summary of the column spread of different cell types in the lobula. Scale bar represents 20 μm.

The following figure supplement is available for figure 6:

**Figure supplement 1.** Layer cross-section views of lobula arbors of the LC neuron types not shown in *Figure 6*.

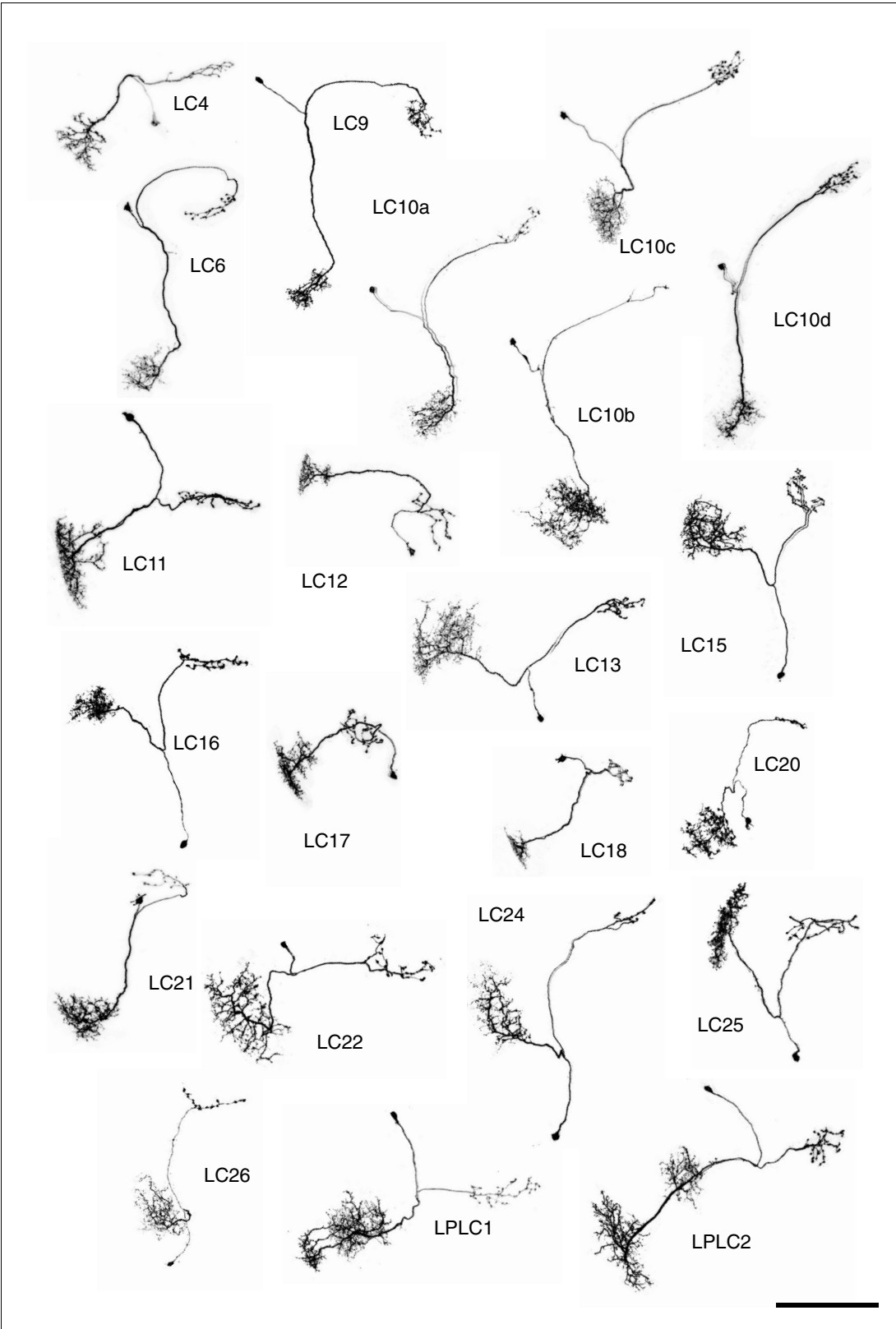

**Figure 7.** Single cell shapes of LC neurons. Maximum intensity projection images of MCFO labeled single cells were manually segmented to exclude other labeled cells or background signal and converted to inverted grayscale images. Cells are shown in a similar orientation (with dorsal approximately up and lateral approximately to the left) and at the same scale. Scale bar represents 50 μm.
*Figure 7 continued on next page*

*Figure 7 continued*

The following figure supplements are available for figure 7:

**Figure supplement 1.** Layer patterns of single cells of 22 LC neuron types.

**Figure supplement 2.** Single cell labeling of LPLC arbors in lobula and lobula plate.

by our driver lines projected to the large medial zone (*Ito et al., 2014*) of the AOTu (*Figure 3G–I*), while inputs to the lateral zone, previously interpreted as LC10C cells (*Otsuna and Ito, 2006*), appear to be projections from the medulla (*Otsuna et al., 2014*; *Panser et al., 2016*). We refer to the subtypes defined here as LC10a-LC10d (lowercase letters) to distinguish them from the previously proposed LC10A-LC10C (*Otsuna and Ito, 2006*). *Costa et al. (2016)* also propose several new LC10 subtypes based on computational clustering of single cell data. These putative cell types are primarily defined by having axonal arbors that are restricted to different positions along the DV axis of the AOTu with corresponding regional patterns of their dendrites along the AP axis of the lobula. Our analyses of retinotopy of LC10 neurons using subtypes specific split-GAL4 lines showed that the axonal terminals of both LC10a (*Figure 4H–J*) and LC10d (*Figure 4K–M*) subtypes are retinotopically distributed along the full DV axis of the AOTu, suggesting that different subtypes form independent retinotopic maps that each cover the entire visual field. For this reason, we believe that the clustering method of Costa et al, which relied on the central brain arbors of LC cells, not their layer patterns in the lobula, led to a misclassification of groups of cells of LC10 neurons at different retinotopic positions as distinct cell types. In agreement with this possibility, attempts to identify GAL4 lines with specific expression in such regionally restricted LC10 subtypes were largely unsuccessful (*Panser et al., 2016*). By contrast, our split-GAL4 driver lines provide genetic support for the LC10 subdivisions we describe here. In sum, within the AOTu the terminals of multiple LC cell types overlap, showing clear retinotopic spatial segregation along one axis.

## LC neuron processes show cell-type specific innervation patterns of lobula layers

Just as the positions of the target glomeruli of LC neurons are indicative of the spatial location of their as yet uncharacterized postsynaptic partners, the distribution of LC neuron arbors across lobula layers can provide clues to the presynaptic inputs to the LC cells. Similar to other optic lobe neuropils, the lobula has a distinctly stratified structure (*Fischbach and Dittrich, 1989*; *Strausfeld, 1976*). To compare layer patterns across samples, we used the anti-Brp reference marker (*Figure 5A*) to both directly identify lobula strata and to enable alignment of different optic lobes to a common reference. In the lobula, anti-Brp staining shows seven bands of alternating labeling intensity (*Figure 5A*); examination of processes of identified neurons with known positions in the lobula indicated that these anti-Brp strata largely correspond to the previously described (*Fischbach and Dittrich, 1989*) lobula layers Lo1 to Lo6 (*Figure 5A* and *Figure 5—figure supplement 1*), with Lo5 represented by two anti-Brp bands of different intensity. Comparison of the layer patterns of different LC cell types, each visualized using a specific split-GAL4 driver line, with the anti-Brp label indicated that, for a given type, layer patterns were similar throughout the lobula (*Figure 5B,C*) but differed for cells of different types (*Figure 5B–D*). In some cases, differences between layer patterns of different LC cell types, though consistent across samples, were small (for example, between LC6 and LC16, LC4 and LC12 or LC25 and LC26) while other pairs of LC cell types occupied, except for connecting neurites, entirely non-overlapping sets of layers (for example, LC4/LC12/LC18 compared to LC25/LC26). In principle, distinct and apparently uniform layer patterns of LC neuron populations (as shown in *Figure 5B–D*) could still consist of multiple cell types with distinct arbor stratifications that were not resolved in these images. However, for nearly all LC neuron types, we found that the layer patterns of individual cells that projected to the same target glomerulus were similar to each other and matched the overall layer patterns seen by labeling the entire cell populations using the type-specific split-GAL4 driver lines (compare LC4 and LC15 patterns in *Figure 5E and D*, respectively; also see *Figure 7—figure supplement 1* for examples of single cell labeling of lobula dendrites for all LC cell types). These results indicate that nearly all optic glomeruli receive input from a

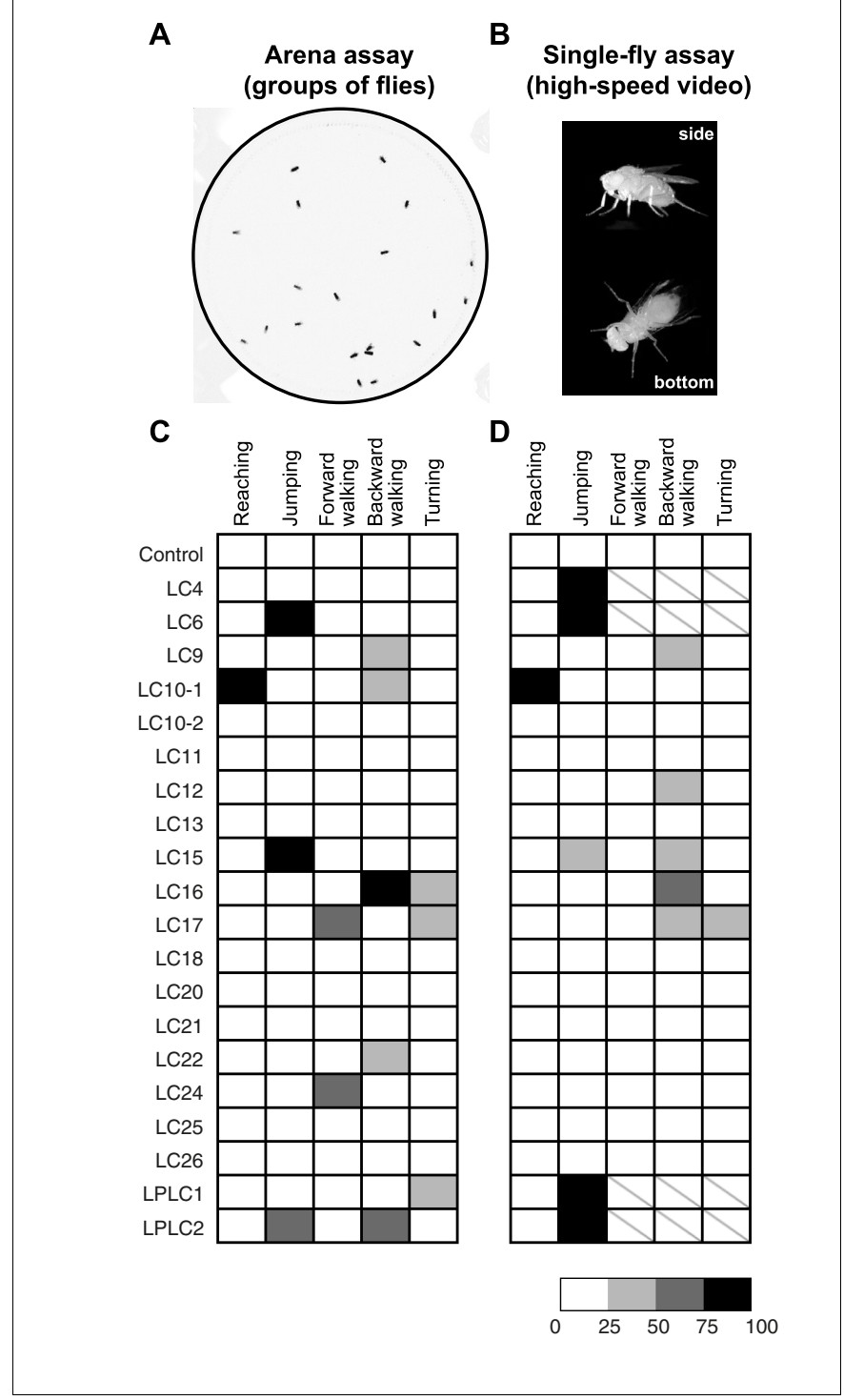

**Figure 8.** Optogenetic activation of LC neurons induces distinct behavioral responses that differ between LC cell types. (**A**) A representative video image of group of freely walking flies in the circular arena assay. (**B**) Representative video images of a freely behaving fly on a small glass platform in the single-fly assay. A side view (upper part of the panel) and a bottom view (lower part) of the fly are simultaneously recorded on a single high-speed video camera with the aid of two small prisms (see Materials and methods). (**C,D**) The results of CsChrimson activation of different LC cell types in the arena (**C**) and the single-fly (**D**) assays are summarized in a grayscale intensity map. Each column represents a distinct behavior and each row represents a different split-GAL4 driver line with a predominant expression in the indicated cell type. Shading represents the behavioral penetrance (percentage of trials or flies of a specific genotype in which a given behavior was observed, see

*Figure 8 continued on next page*

*Figure 8 continued*

***Supplementary file 1B*** for the names of the GAL4 lines used and penetrance values). In both assays, the occurrences of reaching and jumping behaviors were annotated manually, while locomotor behaviors including forward walking, backward walking and turning were determined based on velocity and angular speed derived from automated fly tracking (see ***Figure 8—figure supplement 1***, and Materials and methods). For locomotor behaviors we set a conservative threshold of two standard deviations away from the mean to be considered an activation phenotype (***Figure 8—figure supplement 1E,F***). In this way, we determined the behavioral penetrance for five phenotypes – reaching, jumping, forward walking, backward walking and turning - in both the arena and the single-fly assays. In the single-fly assay, the high jumping penetrance of four LC cell types (LC4, LC6, LPLC1 and LPLC2) resulted in too few flies (<= 12) available for analyses of walking and turning behaviors (indicated with a '\').

The following figure supplements are available for figure 8:

**Figure supplement 1.** Quantification of locomotor behaviors and determination of behavioral penetrance.
**Figure supplement 2.** Variability of LC16 backward walking behavior across trials and across flies.

single LC neuron type that can be recognized independently by either the dendritic arbor stratification in the lobula or the axonal projection pattern.

As mentioned above, an exception to this apparent one-to-one correspondence between LC cell types and target glomeruli were LC10 neurons projecting to the AOTu. The majority of our LC10-specific split-GAL4 drivers are expressed in more than one of the four LC10 subtypes. However, we identified several lines with subtype selectivity, including two lines with expression in a single

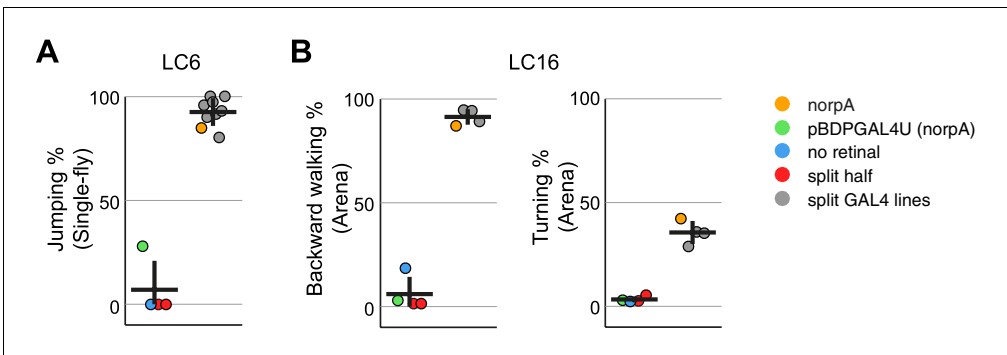

**Figure 9.** Experiments with additional LC6 and LC16 driver lines confirm the activation phenotypes of these cell types. (A,B) Behavioral penetrance for different controls and multiple split-GAL4 driver lines for (**A**) jumping (flies that jumped within 200 ms of stimulation onset) with LC6 controls based on the OL0077B driver line and (**B**) backward walking and turning with LC16 controls based on the OL0046B driver line. Each dot represents an experiment done with a different genotype: orange, LC neuron activation in blind norpA flies that also carry an LC6 (**A**) or LC16 (**B**) split-GAL4 line; green, pBDPGAL4U control in blind norpA flies; blue, flies reared on food without supplemental retinal; red, split-GAL4 DBD or AD halves; grey, genetically distinct split-GAL4 driver lines with targeted expression in LC6 (**A**) or LC16 (**B**). Horizontal and vertical lines indicate mean and standard deviation, respectively, for the control group and split-GAL4 group. The genotypes of the driver lines, behavioral penetrance and total trial and fly counts are listed in ***Supplementary file 1B***. Expression patterns of the split-GAL4 driver lines used are shown in ***Figure 9—figure supplement 1***.
The following figure supplements are available for figure 9:

**Figure supplement 1.** Expression patterns of multiple split-GAL4 driver lines for LC6 and LC16.
**Figure supplement 2.** Quantification of LC16 activation induced locomotor behaviors and determination of behavioral penetrance.

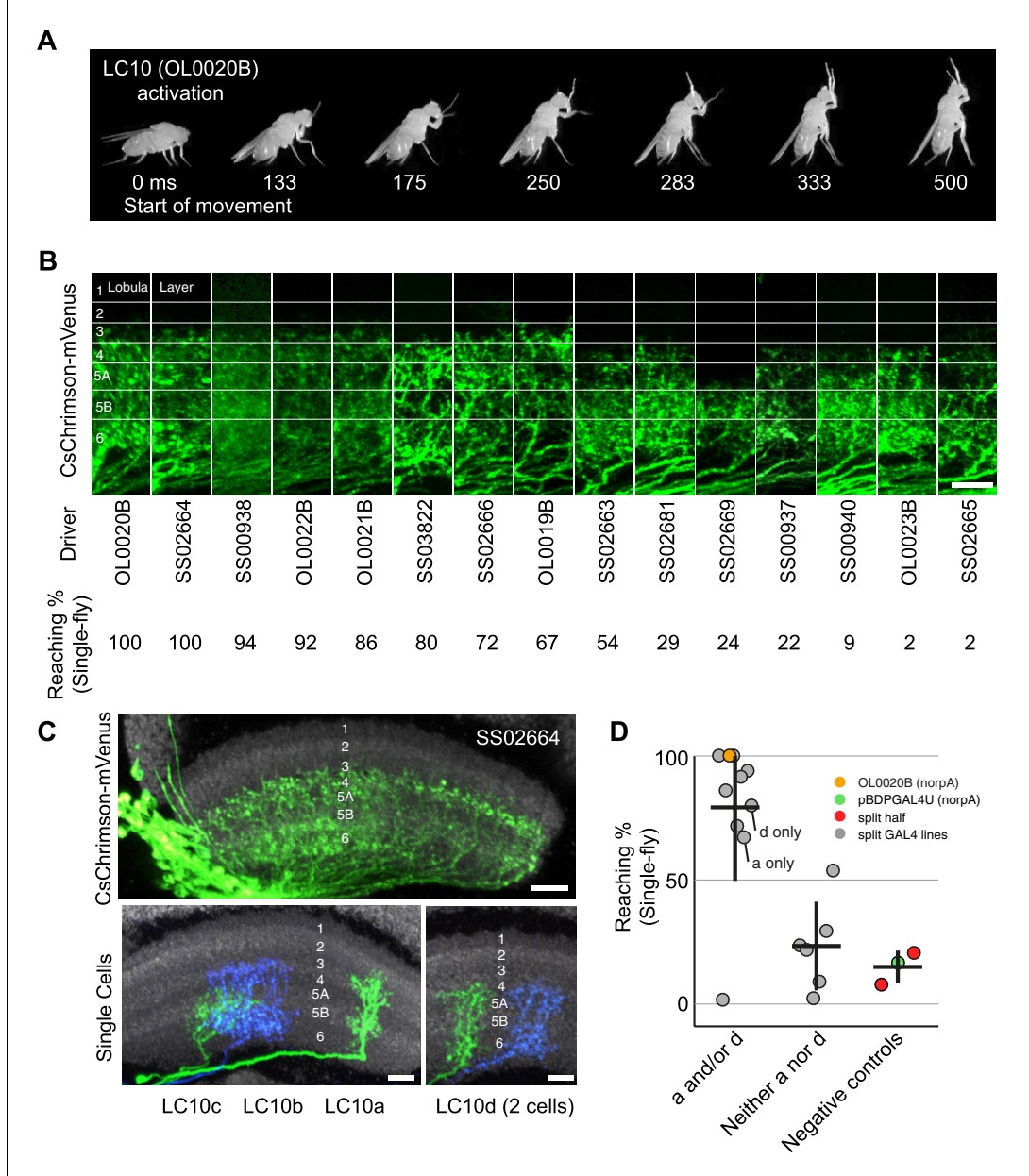

**Figure 10.** Reaching behavior resulting from activation of LC10 subtypes. (**A**) Representative video images of a fly exhibiting reaching behavior in the single-fly assay. Time stamps indicate milliseconds (ms) after the start of reaching. (**B**) Comparison of lobula layer patterns and penetrance of reaching upon optogenetic activation for 15 LC10 split-GAL4 lines. The images show reconstructed views of CsChrimson expression in the lobula (generated using Vaa3D and manually aligned using the anti-Brp reference marker). Approximate layer positions are indicated on the left. Scale bar represents 10 μm. CsChrimson expression using two additional LC10 driver lines (SS00941 and SS00942) resulted in unexpected uncoordinated behaviors in response to optogenetic activation that precluded analyses of reaching behavior. These lines, which are related as they only differ from each other in the insertion site of the AD hemidriver (see **Supplementary file 1B**), were therefore excluded from further analyses. (**C**) Single cell labeling reveals subtype expression patterns of LC10 driver lines. Overall lobula expression of an LC10 split-GAL4 line (SS02664) (top panel; displayed as in **B**) and examples of MCFO labeled single cells from this line (bottom panels) are shown. LC10 subtypes (indicated below the images) of single cells were assigned based on layer pattern, arbor size and shape as follows: LC10c cells mainly arborize in layer Lo5B with some processes in the adjacent layers. LC10a cells also have arbors in Lo5B, but differ from LC10c by having additional processes in the more distal layers Lo2, Lo3 and Lo4. LC10b and LC10d differ from LC10a and LC10c by having major arbors in Lo6 and only few processes in Lo5B. Their distal arbors reach Lo4. LC10b cells are wider than LC10d cells and show many small varicosities (presumably presynaptic sites; compare **Figure 5—figure supplement 2E**) in Lo6. LC10 neurons also differ in their axonal paths in the AOTu: LC10a and LC10d axons run both dorsally and ventrally, and LC10b and LC10c only ventrally (**Figure 3G**; also see **Figure 10—figure supplement 1**). Scale bars represent 10 μm. (**D**) Reaching penetrance observed upon activation of the LC10 split-GAL4 drivers and various control lines in the single-fly assay. Split-GAL4 drivers are grouped by expression patterns in the LC10 subtypes: either subtype a and/or d, or

*Figure 10 continued on next page*

*Figure 10 continued*

neither a nor d. Driver lines were included in the 'a and/or d' category if two or more cells of these types were identified in MCFO experiments (see *Figure 10—figure supplement 2*). Colors representing various controls and split-GAL4 driver lines are the same as those in *Figure 9*.

The following figure supplements are available for figure 10:

**Figure supplement 1.** Expression patterns of LC10 split-GAL4 driver lines.

**Figure supplement 2.** Stochastic single cell labeling reveals LC10 subtype expression patterns of the LC10 split-GAL4 driver lines.

subtype (see *Figure 10—figure supplement 2*). This provides genetic support for the treatment of LC10a, LC10b, LC10c and LC10d as distinct, though related, cell types.

We found that all lobula layers contain processes of at least one LC cell type, although the number of types contributing to each layer varies widely. At one extreme, only LC4 cells have processes in layer Lo1; by contrast, all but three LC (LC4, LC12 and LC18) cell types have processes in layer Lo5. As anatomical overlap between different cells is a necessary condition for synapses between them, the different layer patterns of LC cells can provide clues to their potential connectivity: for example, some LC cell types with Lo5 dendrites might receive inputs from Tm neurons implicated in the processing of spectral information (*Gao et al., 2008*; *Karuppudurai et al., 2014*; *Lin et al., 2016*) (Tm20, shown in *Figure 5—figure supplement 1*, is an example). However, since all lobula layers contain terminals of many neurons, connectivity cannot be inferred based on layer patterns alone, as not all the neuronal types that arborize in a shared layer will be synaptic partners. In addition, the lobula arbors of some LC neurons may be not only postsynaptic but also make some presynaptic contacts; we found that for several LC cell types the presynaptic marker (syt-smHA) used to visualize the target regions was also detectable, though much more weakly, in one or more lobula layers (*Figure 5—figure supplement 2*, *Supplementary file 1A*), suggesting that some LC neurons are also presynaptic in the lobula.

## LC neurons show cell-type specific dendritic arbor sizes and shapes

Like layer patterns, lateral dendritic spread is also a stereotyped characteristic of LC neurons with functional implications. As populations, the dendrites of each of the LC neuron types form retinotopic arrays that cover the entire lobula (see *Figure 1C,F,K*). However, the lateral spread of individual LC neuron arbors in the lobula shows considerable cell-type specific variation (*Figure 6* and *Figure 6—figure supplement 1*). The lateral spread of the lobula arbors of cells of all LC types covered lobula regions corresponding to several of the ~750 visual columns within each eye. The largest arbor spreads were observed for LC11, estimated to be over 60 columns (*Figure 6A* and *Supplementary file 1A*) and LC25, approximately a hundred columns for some cells (*Figure 6—figure supplement 1* and *Supplementary file 1A*). The smallest arbors were those of LC10a (~10 columns; *Figure 6B* and *Supplementary file 1A*). For comparison, some of the medulla inputs to the lobula have lobula arbors corresponding to ~1 visual column (*Figure 6D*) and some lobula tangential cells span all visual columns of the lobula as single cells (*Figure 6E*). Since most LC cell types did not show obvious retinotopy within their target glomeruli, we propose that the distinct arbor spreads of different LC cells determine the spatial extent over which their specific response properties are computed, rather than provide retinotopic information at different spatial resolutions.

Taken together, our estimates of the number of LC cells of each type and their approximate dendritic arbor spread within lobula layers suggest that the dendrites of a given LC type, with the possible exception of the small arbors of LC10a, do not show a strict tiling pattern but rather overlap (*Supplementary file 1A*). We also directly observed overlap of co-labeled cells in MCFO experiments (some examples can be found in *Figure 6—figure supplement 1*) for nearly all LC cell types (*Supplementary file 1A*). Overlap between processes of cells of the same type is also common in the medulla; our previous study (*Nern et al., 2015*) of Dm neurons using similar methods provides detailed examples of both overlapping and strict tiling patterns of medulla neurons.

The morphology of single cells also revealed many additional details of LC neuron arbor structure that appear to be stereotyped within cells of the same type (*Figure 7*, *Figure 7—figure supplement 1*, *Figure 7—figure supplement 2* and *Supplementary file 1A*): For example, the arbor spread,

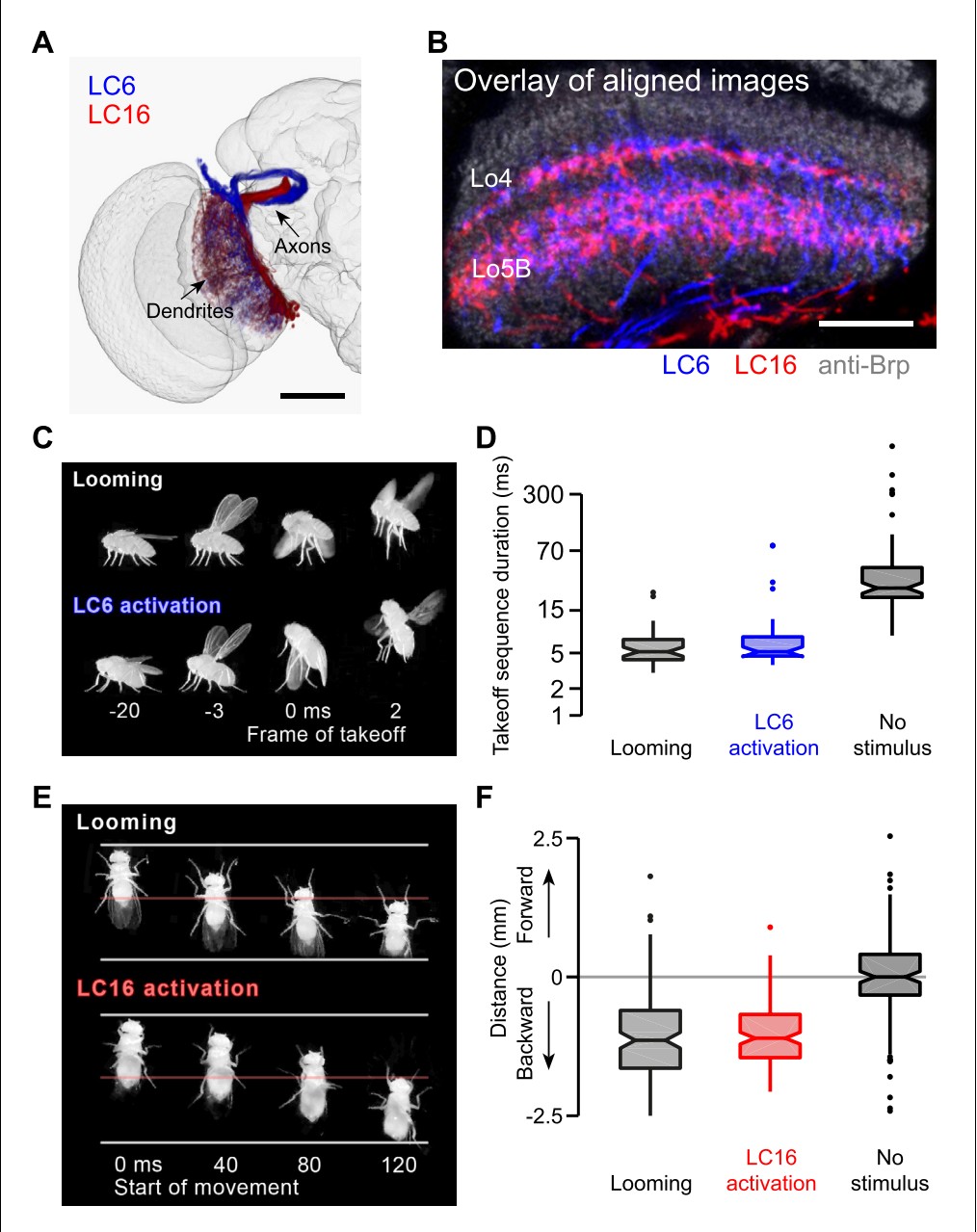

**Figure 11.** LC6 and LC16 activation behaviors resemble avoidance responses evoked by visual looming. (**A**) LC6 and LC16 project to adjacent, non-overlapping target glomeruli. The image was generated using a 3D image rendering software (FluoRender) (**Wan et al., 2012**) on aligned confocal images. (**B**) LC6 and LC16 have similar layer patterns in the lobula. An overlay of substack projections of aligned image stacks is shown. Anti-Brp reference marker is in grey. (**C**) Representative video images from the single-fly assay showing that a looming stimulus and LC6 activation evoke very similar coordinated behavioral sequences, which include wing elevation, middle leg extension and initiation of flight. Time stamp is set at 0 ms for the frame of takeoff. Negative and positive values are for frames before and after takeoff, respectively. (**D**) Notched box plots showing the duration of the takeoff sequence measured as the time from the first moment of wing movement to the last moment of tarsal contact with the ground after the stimulus (Mann-Whitney test, p=0.29 between looming and LC6 activation, and p<0.001 between LC6 activation and no stimulus). (**E**) Representative video images from the single-fly assay showing that a looming stimulus and LC16 activation evoke very similar backward walking behaviors. Time stamp is set at 0 ms for the start of backward walking. (**F**) Total distance flies walked on the platform of the single-fly assay. Positive and negative values are for forward and backward walking, respectively (Mann-Whitney test, p=0.71

*Figure 11 continued on next page*

*Figure 11 continued*

between looming and LC16 activation, and p<0.001 between LC16 activation and no stimulus). Scale bars represent 50 µm (**A**) or 20 µm (**B**).

The following figure supplement is available for figure 11:

**Figure supplement 1.** Behavioral consequences of silencing LC6 and LC16 by Kir2.1 expression.

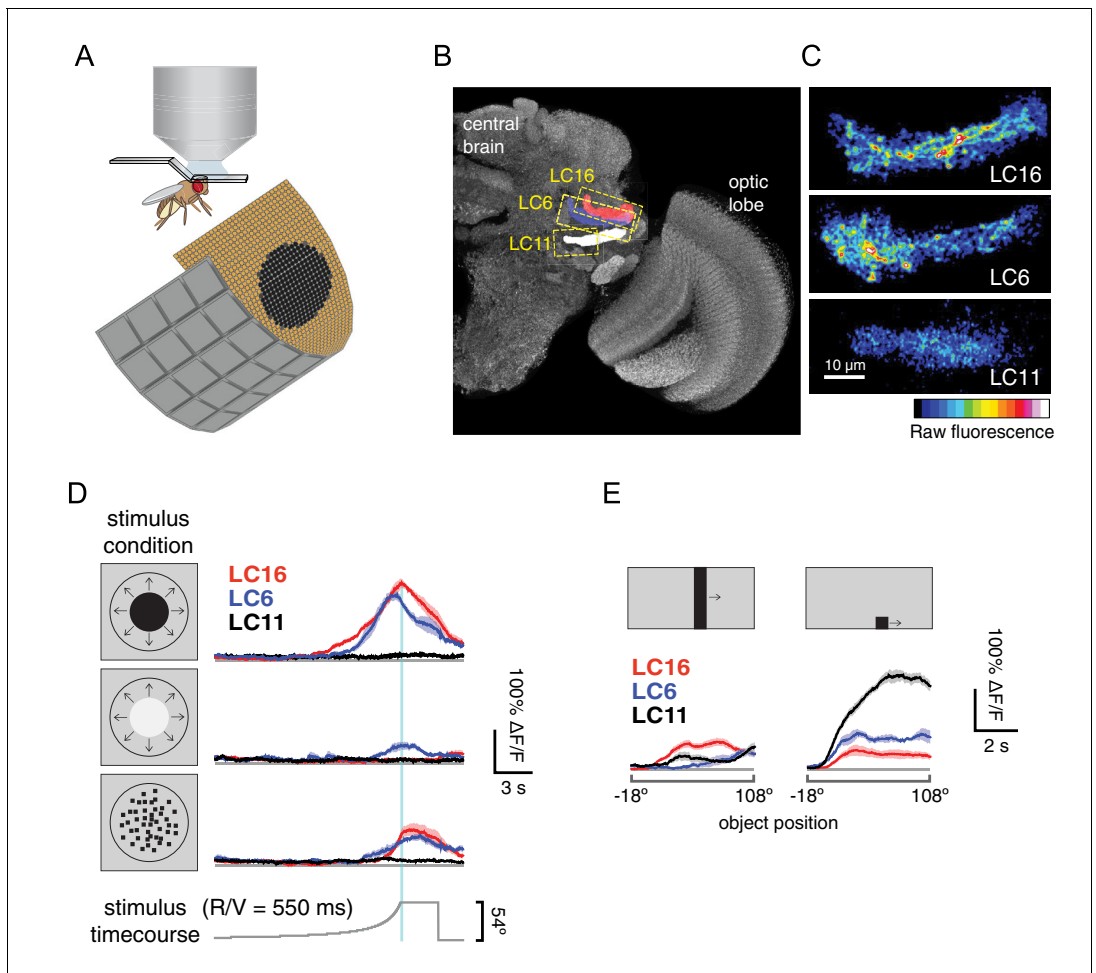

**Figure 12.** LC16 and LC6, but not LC11, respond to visual looming stimuli with robust calcium increases. (**A**) Visual stimuli evoked calcium responses of LC neurons were imaged in head-fixed flies. (**B**) The axon terminals of LC cells bundle to form cell-type specific glomeruli (subset shown in **C**). We imaged from a single glomerulus by using spilt-GAL4 lines labeling individual cell-types (LC16, LC6 or LC11). Representative regions for calcium imaging experiments are marked with the yellow dashed rectangles. Exemplary responses of LC16, LC6 and LC11 to a slow dark looming disk are shown (**C**; each single frame taken from the peak response of an individual fly, distinct genotypes were used to image from each glomerulus). (**D**) LC16, LC6 and LC11 responses to looming visual stimuli are shown for three variants of the stimulus (from top to bottom: dark looming disk, bright looming disk, luminance-matched) expanding at r/v = 550 ms (n = 5 per genotype). Error bars indicate mean ± SEM. Statistics were performed on mean ΔF/F during a time window in which the response peaks (2 s before and after the looming stimulus stops expanding). (**E**) As a comparison to looming stimuli, we also presented moving object stimuli that contain local motion that is distinct from looming. LC11 responds strongly to the motion of the small (9°x9°) spot, but not the long bar (9°x72°) moving object. The objects moved at 22.5°/s, starting 18° left of the visual midline and stopping 108° to the right of the midline. Statistics were performed on mean ΔF/F during the whole stimulus epoch.

The following figure supplement is available for figure 12:

**Figure supplement 1.** LC16 and LC6 are tuned to slower looming speeds.

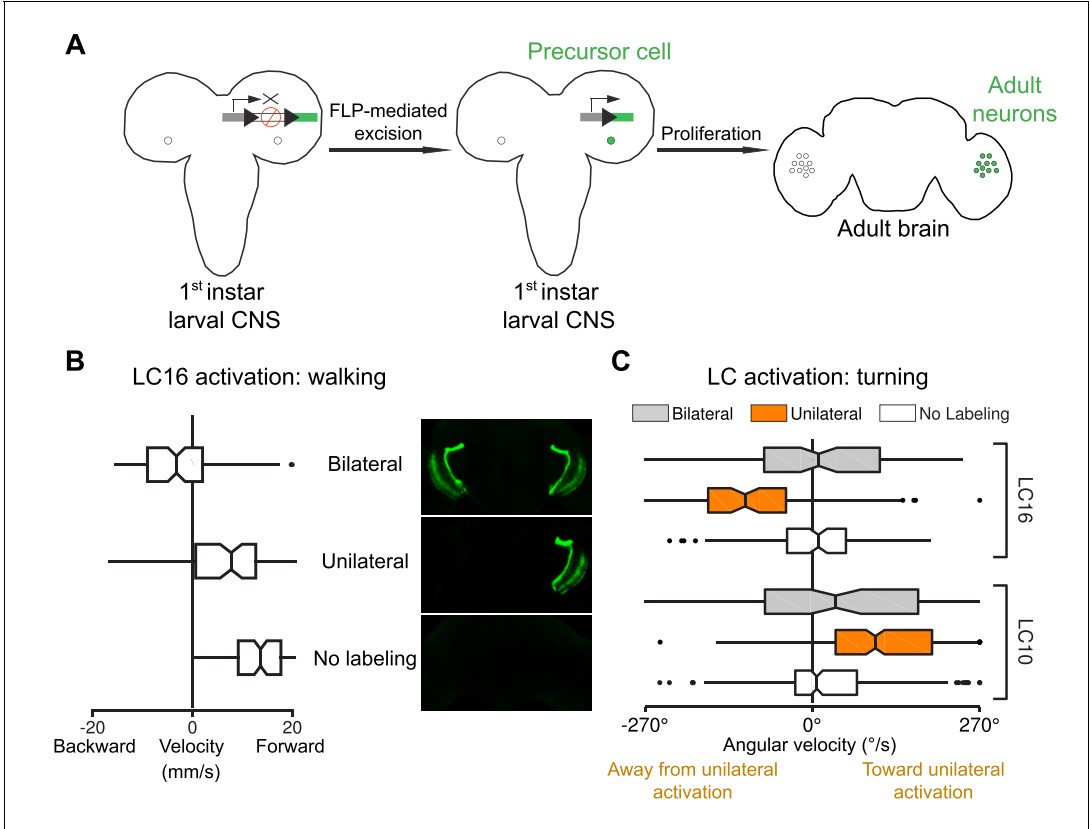

**Figure 13.** Behavioral responses to unilateral LC neuron stimulation differ from bilateral activation behaviors and are directional. (**A**) Schematic illustration of a genetic method for stochastic labeling and activation of LC neurons. A 'stop-cassette' reporter (pJFRC300-20XUAS-FRT>-dSTOP-FRT>-CsChrimson-mVenus in *attP18*) was used for Flp-recombinase mediated control of CsChrimson expression. This reporter/effector construct (small schematic in the right brain hemisphere of the larval brains in the illustration) is based on the 'Flp-out' design (***Struhl and Basler, 1993***). It contains 20 Upstream Activating Sequences (UAS) and a core promoter (grey rectangle) for GAL4-activated expression, a transcriptional terminator (white rectangle with a red prohibition sign) flanked by Flp-recombinase target (FRT) sites (black triangles), and a CsChrimson-mVenus fusion gene (green rectangle). Heat shock induces expression of the Flp-recombinase which can excise the transcriptional terminator, allowing expression of CsChrimson-mVenus under the control of a split-GAL4 driver (not shown). By expressing a limiting amount of Flp-recombinase early in development (first instar larval stage), stochastic stop-cassette excision occurs in LC precursor cells, generating adults in which most or all neurons of one LC neuron type (determined by the split-GAL4 driver) express CsChrimson-mVenus in either no, one, or two optic lobes. (**B**) Strong backward walking behavior requires bilateral LC16 activation. Notched box plots showing the distribution of mean velocity for bilateral LC16 activation (trial count = 101, fly count = 10), unilateral LC16 activation (trial count = 75, fly count = 7) and no labeling controls (trial count = 114, fly count = 11). Behavioral responses of individual flies were assayed and their brains were subsequently dissected to determine expression patterns. Unilateral LC16 activation produced far less backward walking than bilateral activation (Mann-Whitney test, p<0.001). (**C**) Unilateral activation of LC16 and LC10 induces aversive and attractive turning, respectively. Notched box plots showing the distribution of mean angular velocity for different labeling categories of LC16 and LC10: For LC16, trial and fly counts are the same as in (**B**). For LC10, bilateral (trial count = 91, fly count = 8), unilateral (trial count = 268, fly count = 24) and no labeling (trial count = 281, fly count = 22). For unilateral activation, behavioral data from animals with labeling only on the left brain hemisphere were reversed and combined with those from animals with labeling only on the right brain hemisphere. In addition to flies with clear bilateral or unilateral expression, 18 flies in the LC10 stochastic activation experiments had expression in both brain halves but showed differences in the apparent number of labeled LC10 cells between the hemispheres. Because of the wide range of these labeling differences we did not include these flies in the above analysis. However, the behavioral results for this group also showed a turning bias towards the side with stronger labeling (see ***Figure 13—figure supplement 1***), suggesting that even small differences of LC10 activation between the two hemispheres may be sufficient to induce ipsilateral turning.

The following figure supplement is available for figure 13:

**Figure supplement 1.** Turning behavior of LC10 flies with bilateral labeling that is stronger in one brain hemisphere (trial count = 212, fly count = 18).

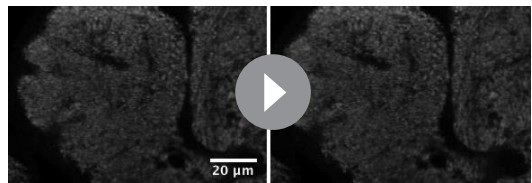

**Video 1.** Composite of aligned images showing a series of sections through the ventrolateral central brain with the target regions of 18 LC (or LPLC) neurons visualized by presynaptic marker expression. Original images, composite image assembly and color scheme are as in *Figure 3*. The anti-Brp reference pattern (grey) of the template brain used for alignment is shown both together with the labeled LC terminals (left panel) and individually (right panel). Terminals of LC10 cell types in the AOTu are not included in the movie. We believe the slight overlap seen between LC12 and LC18 and between LC24 and LC26 is due to imperfect alignment, rather than true overlap.

even for a single cell type, can differ between layers (examples are LC11, LC12, LC17 and LC18) and dendritic processes of some cell types appear to point in a specific direction relative to the main neurite (examples include LC26 and LPLC2).

In summary, the multicolumnar arbor span (*Figure 6* and *Figure 6—figure supplement 1*), intricate arbor shapes (*Figure 7*, *Figure 7—figure supplements 1* and *2*) and diverse and, in most cases, multistratified layer patterns (*Figure 5*, *Figure 7—figure supplements 1* and *2*) of LC neurons, suggest that these neurons spatially integrate inputs from multiple presynaptic cells of several types. These anatomical characteristics suggest that as a whole, the population of LC cell types has the potential to encode a diversity of visual stimuli, with large differences expected between distinct types.

## Optogenetic activation of LC neurons results in distinct behavioral responses

What information do LC neurons provide to the central brain? As discussed in the Introduction, the anatomical reorganization resulting from the convergence of LC axons into largely cell-type specific glomeruli, has led to the speculation (*Mu et al., 2012*; *Strausfeld and Okamura, 2007*) that the activity of members of a given LC cell type primarily signals the presence of a particular behaviorally relevant visual feature rather than its precise location. Thus, it is reasonable to expect that strong artificial activation of an individual LC neuronal cell type might elicit a behavioral response that could provide clues to the function

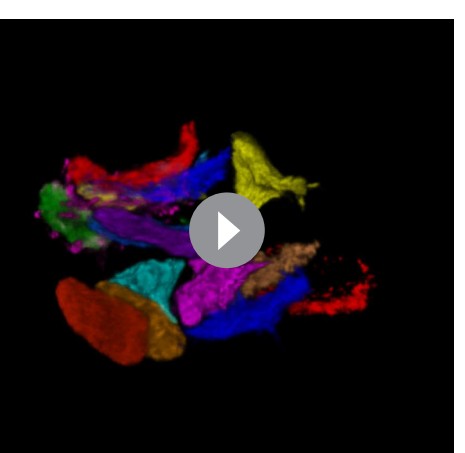

**Video 2.** Three-dimensional rendering of the target regions of LC (or LPLC) neurons in the ventrolateral central brain shown in *Figure 3* and *Video 1*. Image assembly and color scheme are as described in *Figure 3*. Anti-Brp reference pattern is in grey. Terminals of LC10 cell types in the AOTu are not included. The movie was generated using Vaa3D.

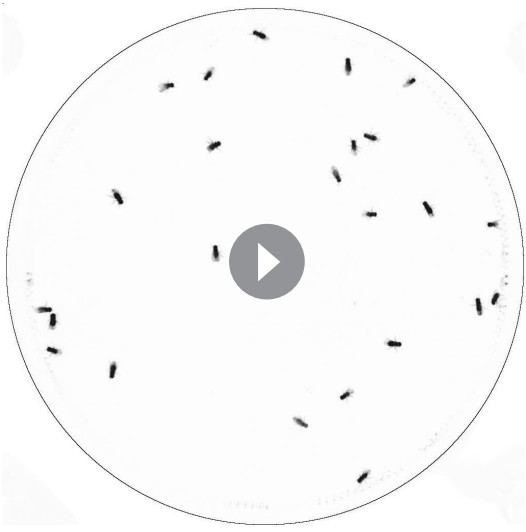

**Video 3.** An example of an LC neuron activation phenotype (backward walking and turning) in the arena assay. A representative video showing groups of freely walking flies tested in the circular arena. The video is shown at 0.4x actual speed and the red indicators at the corners indicate the timing of optogenetic activation (1 s). This example shows a highly penetrant backward walking and turning phenotype resulting from LC16 activation.

of these neurons. To explore this possibility, we conducted an optogenetic activation screen, using our LC cell-type specific split-GAL4 driver lines to genetically target expression of CsChrimson, a red light-activated cation channel (*Klapoetke et al., 2014*), to specific LC neuron populations.

We initially focused on 20 split-GAL4 driver lines, each with expression in a different LC neuron type. For LC10 cells, we chose two driver lines that together included all four LC10 subtypes described above. Driver line expression patterns were confirmed by imaging CsChrimson-mVenus expression in flies of the same genotypes used in the behavioral experiments (*Figure 2* and *Figure 2—figure supplement 1*). We transiently depolarized these cells with a short pulse of red light in the context of two independent, complementary behavioral assays (*Figure 8A,B*). One assay used a circular arena (100 mm diameter and 3 mm high) (*Aso et al., 2014b*; *Klapoetke et al., 2014*), in which groups of about 25 freely walking flies (*Figure 8A* and *Video 3*) were subjected to periodic, repeated light pulses. The second was a single-fly assay (*Figure 8B*, see Materials and methods) in which flies were released individually onto a small (5 mm$^2$) open platform where they were exposed to light activation and their response recorded in two views (side and bottom) at high temporal and spatial resolution, permitting analysis of behaviors with greater detail. All activation experiments were carried out under infrared light to permit video recording while minimizing potential activation of CsChrimson by ambient light.

An overview of the screen results is presented in *Figure 8C,D* and in *Supplementary file 1B*. The table in *Supplementary file 1B* includes some additional behavioral responses that were below the 25% penetrance threshold used in *Figure 8C,D*. The red light pulse used for optogenetic activation itself induced weak behavioral responses in flies from a control driver line without detectable expression in the nervous system (the 'enhancerless' pBDPGAL4U, [*Pfeiffer et al., 2010*]). These control responses (also see *Klapoetke et al. [2014]*) included a slight increase in forward walking speed or turning. However, we found that red light activation of CsChrimson in LC neurons resulted in several phenotypes that were dramatically different from these baseline responses (*Videos 3–6*). These phenotypes included larger increases in forward walking speed or turning activity as well as three striking behaviors rarely seen in control flies, which we refer to as jumping, reaching and backward walking.

We found that the relationship between LC neuron type activated and observed behavior was not one-to-one. In some cases, the same behavior was elicited from activation of different LC cell types. For example, five different cell types (LC4, LC6, LC15, LPLC1 and LPLC2) drove highly penetrant jumping in at least one of the two assays. In other cases, multiple behaviors were observed from activation of a single LC neuron type. For example, LPLC2 activation elicited both jumping and backward walking behaviors with about equal penetrance in the arena assay. Overall, when activated, nearly half of LC neuron types (10/22) drove one of the five behaviors we assessed in a majority of flies tested. Since we did not independently confirm CsChrimson stimulation of LC neurons in a separate assay, we cannot exclude the possibility that the absence of a clear behavioral response of some LC cell types may have been simply due to insufficient activation of these neurons under our experimental conditions.

The results obtained with the two assays were largely consistent but did show some differences. For example, the jumping phenotype of the LC4 and LPLC1 lines had much higher penetrance in the single-fly assay than in the arena assay whereas the opposite was the case for the forward walking phenotype of the LC24 split-GAL4 driver (*Figure 8C,D*). We attribute these differences to design differences between the

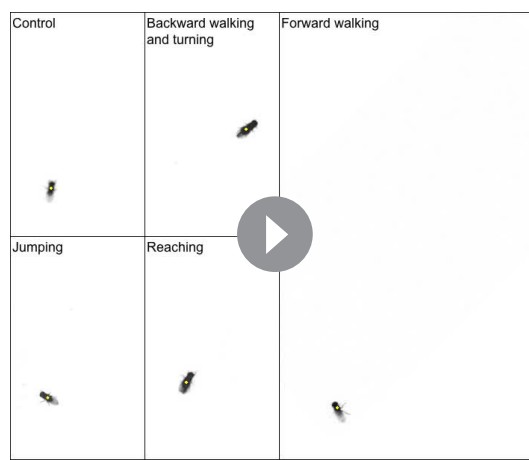

**Video 4.** Examples of distinct LC neuron activation phenotypes in the arena assay. One representative fly for each phenotype is shown before, during and after optogenetic stimulation (1 s each, 3 s in total). Flies' centers of mass are tracked and their trajectories are represented with blue (before and after stimulation) or red (during stimulation) dotted lines. The video is shown at 0.4x actual speed.

two assays; in particular, the higher intensity and shorter duration of the red light stimulus used in the single-fly assay (50 ms pulse of 3 mW/mm$^2$ compared to the 1 s pulse of 94 μW/mm$^2$ stimulus in the arena) and the smaller area of the single-fly assay platform, which limits assessment of walking and turning behaviors.

The screen results suggest that CsChrimson-mediated acute depolarization of individual LC cell types can induce diverse, cell-type specific behavioral responses. To confirm these phenotypes and pursue a more detailed analysis, we selected three LC neuron driver lines that produced robust and highly penetrant activation phenotypes in both assays: LC6 (jumping), LC10 (reaching) and LC16 (backward walking) (*Figure 8C,D*, *Videos 3–6*). We used the higher resolution, single-fly assay to further resolve details of LC6 and LC10 phenotypes. LC16 phenotypes, which also include a distinct turning component (*Figure 8—figure supplement 1A–D*), were further examined in the larger arena assay.

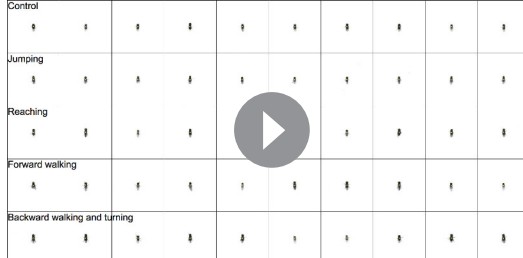

**Video 5.** Examples of distinct LC neuron activation phenotypes in the arena assay. 10 representative flies for each phenotype are shown for the duration of optogenetic stimulation (1 s). Flies' centers of mass are tracked for the duration of stimulation and their trajectories are represented with red dotted lines. Many flies with the reaching behavior also extended one or both wings in response to CsChrimson activation, though we did not further characterize this aspect of the behavior in this study. The video is shown at 0.2x actual speed.

To confirm that the jumping, backward walking, and turning phenotypes could not be attributed to the light response or unrelated differences in genetic background, we performed further control experiments with LC6 and LC16 lines. We combined our split-GAL4 driver lines with a *norpA* mutation that renders the fly blind. CsChrimson activation of these lines produced the same behaviors as seen in non-*norpA* flies (*Figure 9* and *Supplementary file 1B*), indicating that the behavior is not triggered by the light itself (behavioral penetrance measured in genetically blind flies and experimental lines were not significantly different, binomial test, p=0.06 for jumping, p=0.10 for backward walking and p=0.40 for turning). Additionally, parental lines that do not express CsChrimson as well as flies reared on food without supplemental retinal show little or no jumping, backward walking or turning in response to red light (mean penetrance of 0% for jumping, 7% for backward walking and 3% for turning) (*Figure 9*, *Figure 9—figure supplement 2* and *Supplementary file 1B*). We also tested several additional, genetically different, split-GAL4 driver lines with targeted LC6 or LC16 expression (*Figure 9—figure supplement 1*) and found that these lines produced the same, highly penetrant jumping or backward walking and turning behaviors (*Figure 9*, *Figure 9—figure supplement 2* and *Supplementary file 1B*; 80–100% penetrance for jumping, 89–95% for backward walking and 29–36% for turning). These results confirm that the distinct LC6 and LC16 phenotypes observed in the screen are due to the experimental activation of these cell types. Furthermore, an independent thermogenetic activation screen of more than 2000 lines from the Janelia GAL4 collection also identified the LC6 and LC16 cell types as producing the jumping and backward walking phenotypes through correlation between behavior and anatomy (A A Robie and

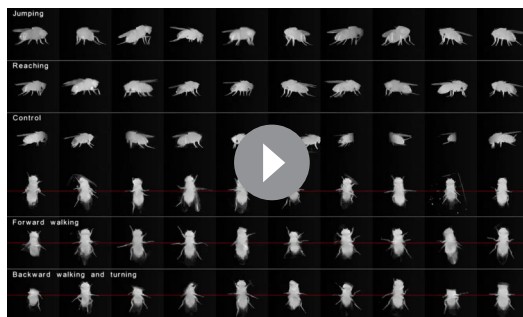

**Video 6.** Examples of distinct CsChrimson activation phenotypes in the single-fly assay. 10 representative flies of each genotype are shown during the 50 ms optogenetic stimulation and for the following 450 ms. Jumping and reaching phenotypes are shown in the side view whereas forward walking, backward walking and turning phenotypes are shown in the bottom view. The pBDPGAL4U control flies are shown in both the side and bottom views. In the bottom view, a red horizontal line through flies' center of mass is used to help visualize forward walking, backward walking and turning behaviors.

K Branson, personal communication, September 2016). Thus, activation of specific LC neuron types can result in highly penetrant, cell-type specific behavioral responses.

## LC10 subtypes have different activation behaviors

Activation of LC10 can result in a reaching behavior (*Figure 10A* and *Video 6*). However, the two LC10 split-GAL4 driver lines we used in our screen produced different activation phenotypes (*Figure 8C,D* and *Supplementary file 1B*). Unlike other LC cell types, LC10 can be divided into four subtypes that each has a distinct layer pattern in the lobula (*Figure 5F*), while projecting to the same target glomerulus (*Figure 3G–I*). These different lobula layer patterns suggest different presynaptic inputs and, as a result, different visual response properties of LC10 subtypes that may be associated with different behavioral outputs. To confirm the reaching behavior observed in the screen with additional LC10 driver lines and to look for possible correlations between subtype expression and this behavior, we assayed a panel of 15 LC10 split-GAL4 driver lines (*Figure 10*; *Figure 10—figure supplement 1*). Upon activation, eight LC10 lines showed strong reaching responses (>60%) and six driver lines showed medium or little reaching (<30%) (*Figure 10B*). Consistent with possible functional differences between LC10 subtypes in these experiments, we found that the expression patterns of the lines with the strongest reaching behavior appeared to differ from those of the other lines by having denser processes in more distal lobula layers (mainly ~ Lo3 and Lo4) (*Figure 10B*).

To determine which of the four LC10 subtypes were included in each of the 15 lines used in our behavioral assay, we visualized individual LC10 cells by MCFO labeling (*Figure 10C*, *Figure 10—figure supplement 2*). Although individual LC10 cells showed considerable morphological variability, nearly all labeled LC10 cells in these lines could be readily classified as belonging to one of the four subtypes. This allowed us to determine the presence of LC10a, b, c or d in each driver line. We found LC10a and/or LC10d cells in all lines with strong reaching phenotypes (*Figure 10C,D* and *Figure 10—figure supplement 2*). These lines included one split-GAL4 driver (OL0019B, *Figure 10D* 'a only') that appeared specific for the LC10a type and another driver (SS03822, *Figure 10D* 'd only') specific for the LC10d type. Control experiments confirmed that the observed reaching behavior did not result from a simple response to light or from unrelated differences in genetic background (*Figure 10D* and *Supplementary file 1B*). Thus, activation of LC10a or LC10d neurons alone is sufficient to drive highly penetrant reaching behavior. The lines with weaker reaching phenotypes showed labeling of LC10b and LC10c cells and little or no expression in the other two LC10 subtypes, further suggesting that, unlike LC10a or LC10d, LC10b and LC10c neurons only have a minor, if any, role in generating the reaching behavior.

## LC6 and LC16 activation induce responses that resemble avoidance behaviors

The LC10 subtypes examined above have diverse layer patterns in the lobula but project to a common target region (*Figure 5F* and *Figure 3G–I*). By contrast, LC6 and LC16 are examples of LC cell types that have very similar arbor stratification in the lobula, but project to distinct optic glomeruli (*Figure 11A,B*, also see *Figures 3*, *5* and *Video 1* and *2*). Despite the similarity in their lobula arbors, LC6 and LC16 show different, highly penetrant, activation-induced behaviors—jumping and backward walking, respectively (*Figure 11B* and *Videos 4–6*). The 'jumping' phenotype of LC6 is reminiscent of fly takeoff, a behavior that occurs both spontaneously and in response to specific visual stimuli such as a predator-mimicking loom (*Card and Dickinson, 2008b*). To further explore this similarity, we made use of the single-fly assay that provides a platform in which optogenetically triggered behaviors, such as LC6-mediated jumping, and responses to specific visual stimuli can be directly compared in identical experimental conditions. To visually elicit takeoff behavior, we presented flies with a looming stimulus previously shown to mimic a predator's approach and trigger fast escape responses (*von Reyn et al., 2014*). We compared high-speed videos of these flies to similar recordings of the jumping phenotype resulting from optogenetic depolarization of LC6. For the flies that took off in response to the looming stimulus at 90° azimuth (47/174 flies) we observed a consistent, coordinated behavioral sequence that started with the fly beginning to elevate its wings, followed by rapid middle (jumping) leg extension and initiation of flight as previously described (*Figure 11C*) (*Card and Dickinson, 2008b*; *von Reyn et al., 2014*). Flies perform the same sequence when taking off voluntarily, however the elevation of their wings is slower (*Card and*

*Dickinson, 2008a*), resulting in a significantly longer-duration takeoff sequence than those evoked by looming stimuli. Strikingly, the CsChrimson-mediated LC6 depolarization not only produced a takeoff with this same sequence of events (*Figure 11C*), but the duration of the takeoff sequence was more similar to that of a looming-evoked takeoff than a voluntary one performed when no stimulus was present (*Figure 11D*). Thus, LC6 depolarization results in a behavior very closely resembling a looming-evoked escape response.

In another set of experiments, we found that backward walking is also a possible response to a specific type of looming stimulus. When a fast-approaching looming stimulus was presented in front of the fly at 0° azimuth, many of the flies that did not take off during the observation period instead responded with a backward walking behavior (174/200 flies) similar to that induced by optogenetic depolarization of LC16 (*Figure 11E*). It has been shown that backward walking can also be part of a fly's response to a predator. For example, upon close encounter with a nymphal praying mantis, wild-caught *Drosophila melanogaster* flies have been reported to occasionally respond with a backward walking or 'retreat' behavior usually followed by turning (*Parigi et al., 2014*). We found that looming and LC16 depolarization both caused flies to move backward, and they do so with similar speeds. In contrast, without a visual stimulus, flies on the platform moved less, and their average movements were in a forward direction (*Figure 11F*). These comparisons show that LC6 and LC16 activation phenotypes resemble typical, visually guided behaviors.

In view of these findings, we performed additional experiments to ask whether silencing of LC6 or LC16 reduces the amount of takeoff or backward walking in response to visual stimuli. We presented the same looming stimuli used in *Figure 11* to flies that expressed the inward-rectifying potassium channel Kir2.1 in LC6 or LC16 neurons and to control flies (*Figure 11—figure supplement 1*). Neither cell type appeared to be essential for the assayed behaviors: No reduction of backward walking compared to controls was observed for LC16 (*Figure 11—figure supplement 1C*) and a reduced jump frequency when blocking LC6 was apparent only for the fast loom (*Figure 11—figure supplement 1A*). However, no difference in takeoff sequence duration was observed (*Figure 11—figure supplement 1B*). Because multiple LC neuron types show jumping or backward walking responses to optogenetic stimulation (*Figure 8*) and non-LC cell types have also been reported to contribute to looming-evoked escape (*de Vries and Clandinin, 2012*), the simplest explanation of these results is that functional redundancy exists at the level of neurons mediating looming-evoked escape.

## LC neurons selectively encode specific visual features

To explore whether LC6 and LC16 encode visual features, such as looming, that are sufficient to evoke jumping and backward walking (*Figure 11*), we investigated the visual responses of these cell types using in vivo two-photon Ca2$^+$ imaging from head-fixed flies (*Figure 12A*). We measured calcium responses of single LC neuron types by imaging from the axons within each glomerulus using an imaging plane selected to obtain the largest slice through the volume of the glomerulus (*Figure 12B*). In agreement with the anatomical results showing that retinotopy is not simply preserved in the LC6 and LC16 glomeruli (*Figure 4A–C*, *Figure 4—figure supplement 1A–C*), we observed that looming stimuli evoke responses in several axons that span each glomerulus (*Figure 12C*). We quantified the population response of these axons by integrating the calcium signals within the glomerulus region.

Calcium signals integrated over each of the LC6 and LC16 glomeruli both show similar increases in response to dark looming disks (*Figure 12D*, top row). Both cell types appear similarly tuned, responding with larger calcium increments to the slower looming speeds presented (*Figure 12—figure supplement 1*, top row). Both cell types are selective to dark looming stimuli as a looming disk that was brighter than the background did not elicit large responses (*Figure 12D* and *Figure 12—figure supplement 1*, middle row). Looming stimuli provide compound visual cues, so to further test for the specificity of these neurons' responses to dark looming objects, we presented a luminance-matched stimulus that darkened over time with the same temporal profile as the dark looming disk, but lacked any coherent edge motion. This stimulus elicited moderate responses in both cell types that were significantly smaller than the dark looming disk (*Figure 12D* and *Figure 12—figure supplement 1*, bottom row; Mann-Whitney test, p<0.01 for both LC6 and LC16), indicating their selectivity for stimuli with looming motion.

To confirm that the similar responses of LC6 and LC16 to looming stimuli were a genuine property of these cells, we performed the identical experiments on an additional LC neuron type. We measured the responses of LC11 (*Figure 12B,C*), that was selected because its dendrites arborize in lobula layers that are distinct from LC6 and LC16 (*Figure 5* and *Figure 7—figure supplement 1*), and because LC11 was not found to have any strong activation phenotypes across our behavioral assays (*Figure 8C,D*). LC11 did not show calcium response changes to any of the looming-related stimuli (*Figure 12D* and *Figure 12—figure supplement 1*, black trace). However, we observed large responses from LC11, when we presented simpler moving stimuli that did not contain looming motion. LC11 showed large responses to a small moving object, and more moderate responses to a moving bar spanning our visual display (*Figure 12E*). In contrast, LC6 and LC16 also showed calcium responses to the small object, but these responses were smaller than those to the loom stimuli, and much smaller than those of LC11 (*Figure 12E*, Mann-Whitney test, p<0.01 for both LC6 and LC16). Taken together, these results demonstrate that LC6 and LC16 exhibit selectivity for slow, dark looming objects, while LC11 encodes a distinct set of visual features.

## Behavioral responses to unilateral LC neuron activation

Our results support the idea that LC cell types respond to the presence of specific visual features in the environment and that the activation of several LC neuron types can simulate the presence of such features, triggering appropriate behavioral responses. However, during most natural visually guided behaviors, flies respond not only to the general presence of a particular visual feature but also to its location. For example, flies use visual cues to orient toward the far side of a gap when attempting to cross it (*Pick and Strauss, 2005*) or toward another fly during courtship behavior (*Coen et al., 2016*). Flies also adjust their takeoff direction relative to the azimuthal orientation of a looming stimulus (*Card and Dickinson, 2008b*) and show a range of orientation behaviors that depend on the location of objects (*Aptekar et al., 2012*; *Bahl et al., 2013*; *Heisenberg and Wolf, 1984*; *Maimon et al., 2008*; *Reiser and Dickinson, 2010*). Such orientation behaviors must depend on neural circuits that convey the spatial location of specific visual features. Indeed, simultaneous activation of most or all LC neurons of a particular type, as in the experiments described above, is probably rather uncommon under natural conditions, and may partially explain why our activation screen revealed robust phenotypes for only half of the LC cell types. Taking LC11 as an illustrative example, if a function of this population is to signal the presence of a small moving object to downstream circuits, then it is not surprising that activating all LC11 neurons simultaneously failed to result in a coherent behavioral response. While our anatomical data suggest that most LC neurons largely discard retinotopic information, the inputs from the fly's two eyes remain distinct.

In order to provide optogenetic stimuli that, similar to many naturalistic visual stimuli, differ between the two eyes, we designed a genetic approach to activate LC neurons on only one side of the brain. By using a construct in which GAL4-driven CsChrimson expression requires the prior removal of a transcriptional terminator, we can add temporal control to the cell-type specificity provided by the split-GAL4 driver (*Figure 13A*, and Materials and methods). If excision of the transcriptional terminator is induced for a brief time window early in development—after the precursor populations for LC cells in the left and right optic lobes have been established, but before extensive cell proliferation has occurred—animals can be readily obtained in which cells specific to that GAL4 driver on only one side of the animal are labeled. As a stochastic method, this approach also generates flies with bilateral labeling or no labeling (*Figure 13B*) providing both positive and

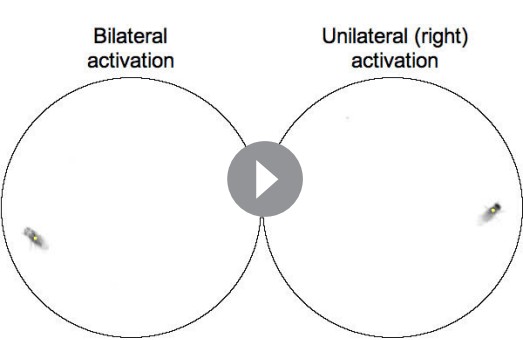

**Video 7.** Examples of phenotypes upon bilateral and unilateral activation of LC16 in the stochastic activation experiment. A representative fly for bilateral and unilateral LC16 activation is shown before, during and after optogenetic stimulation (1 s each, 3 s in total). Flies' centers of mass are tracked and their trajectories are represented with blue (before and after stimulation) or red (during stimulation) dotted lines. The video is shown at 0.4x actual speed.

negative controls of identical genotype and experimental history. To correlate behavioral responses with expression patterns, individual flies were tested in a modified arena (see Materials and methods) and then retrieved to examine labeling patterns by histology.

We applied this method to two LC neuron populations, one of whose activation induces an avoidance response, LC16, and the other an approach by reaching, LC10. Expression patterns were scored as bilateral, unilateral (on the left or right), or no labeling (*Figure 13B*). The behaviors of flies with either bilateral or no expression were consistent with what we observed in our earlier experiments (*Figure 8* and *9*). Unlike bilateral activation, unilateral LC16 optogenetic depolarization only rarely resulted in backward walking (*Figure 13B*), but instead generated strong turning responses (*Figure 13C* and *Video 7*). This turning behavior had a strong directional bias with turns predominantly away from the activated side (*Figure 13C*), suggesting a directional avoidance response. A possible interpretation of these results is that bilateral activation might represent a large object directly ahead, resulting in backward walking, while unilateral activation would mimic an object on one side of the fly, resulting in turning towards the opposite side. Interestingly, unilateral activation of LC10 also caused a strong turning bias, but in the opposite direction; flies predominantly made turns toward the activated side and the most common LC10 activation phenotypes were now turning responses rather than reaching (*Figure 13C*). Taken together, these data show that (1) bilateral and unilateral activation of LC neurons can have different behavioral consequences and (2) unilateral activation can result in either attractive or aversive turning behaviors, depending on the cell type. These results suggest that LC neurons can convey information on both the nature of a visual feature, by means of the differential activity of different LC cell types, and its location, by the differential activation of the same type in the two optic lobes.

## Discussion

In this report, we present anatomical and functional studies of lobula columnar (LC) cells, prominent visual projection neurons from the lobula to target regions in the central brain called optic glomeruli. Comprehensive anatomical analyses of the dendritic arbors and central brain projections of LC neurons support the notion that these cells encode diverse visual stimuli, distinct for each LC cell type, and convey this information to cell-type specific downstream circuits. Precise genetic tools that target individual LC cell types allowed us to explore the behavioral consequences of optogenetic activation of these cell types. We found that activating cells of single LC neuron types was often sufficient to evoke a range of coordinated behaviors in freely behaving flies. Using two-photon calcium imaging from head-fixed flies, we showed that two LC cell types with activation phenotypes similar to avoidance responses, selectively encode visual looming, a stimulus that also evokes similar avoidance behaviors, while a third cell type responded strongly to a small moving object. These results suggest that LC cell types encode visual features that are relevant for specific behaviors. Activation of LC cells in only one brain hemisphere can result in either an attractive or repulsive directional turning response, depending on cell type. Thus which LC neuron channel is activated determines the valence of the behavior, whereas comparison across the brain by two such channels of the same type provides information about the location of relevant visual features.

### Distinct stratification of LC neuron dendrites and selective visual responses suggest LC cells encode diverse, cell-type specific visual stimuli

Anatomical properties of LC neurons have been previously described both in *Drosophila* and other Diptera (*Fischbach and Dittrich, 1989*; *Otsuna and Ito, 2006*; *Strausfeld and Okamura, 2007*). Our work extends these studies by providing a comprehensive description of LC neurons in *Drosophila*, including the identification of several previously unreported cell types. Further, we combine these anatomical analyses with the generation of highly specific genetic markers (split-GAL4 lines) for each cell type. We found that each of the 22 LC types described here has morphologically distinct dendritic arbors in the lobula with stereotyped arbor stratification, size and shape. As observed in the medulla, where synapse-level connectomics data are available for many cell types (*Takemura et al., 2013*), different layer patterns and arbor shapes are likely to reflect differences in synaptic connectivity and neuronal computation. Arbors of LC neurons are found in all lobula strata, though with large differences between layers. Only LC4 (and perhaps LPLC1 and LPLC2) cells are

potentially postsynaptic to neurons in the most distal lobula layer, Lo1, while other strata such as Lo4 and Lo5B include processes of more than half of the LC types. The presence of at least some LC dendrites in each lobula layer implies that all of the about 50 different interneuron types that convey visual information from the medulla, and to a lesser extent from the lobula plate, to the lobula, are potentially presynaptic to some LC cells, although a far smaller number is likely presynaptic to any single LC cell type. The predicted differences in the synaptic inputs to different LC cell types also suggest that they will differ in their responses to visual stimuli. Thus, individual LC neuron types are expected to encode specific visual stimuli, while the population of all LC cell types together should signal a wide range of behaviorally relevant visual features.

The visual responses of several LC cell types measured using two-photon calcium imaging (*Figure 12*) support the expectation that different types selectively respond to different visual features. The three LC neuron types examined preferentially responded to distinct stimuli, with either a dark looming stimulus (LC6 and LC16) or a small moving object (LC11) evoking the strongest measured responses. LC6 and LC16 showed stronger responses to a dark expanding disc than to related stimuli such as an expanding bright disk or a darkening stimulus that lacks the expanding motion. The reduction in the LC6 and LC16 responses when the edge motion is removed from the stimulus is precisely what is expected of loom-sensitive neurons and is reminiscent of behavioral studies in houseflies showing that darkening contrast combined with edge motion is the most effective stimulus for triggering takeoffs (*Holmqvist and Srinivasan, 1991*). Consistent with their similar responses in the imaging experiments, LC6 and LC16 have very similar lobula layer patterns while LC11 has a different arbor stratification indicating that LC11 receives inputs from a different set of medulla cell types than LC6 and LC16.

It is likely that the selectivity for visual stimuli observed in LC neuron responses is both a property of the stimulus selectivity of their inputs—some selectivity was seen while imaging in the dendrites of a few LC cell types (*Aptekar et al., 2015*)—and specific computations implemented by individual LC neuron types. In addition, cells post-synaptic to the LC cells may integrate the responses of several individual LC neurons of the same type to provide more robust detection of specific visual features. For example, while LC6 and LC16 cells as populations are strongly excited by dark looming stimuli, we currently do not know whether individual LC6 and LC16 neurons, which have dendritic extents well below the maximum size of our looming stimuli, and also well below the size known to elicit maximal behavioral responses (*von Reyn et al., 2014*), show the same response properties. Our anatomical data and genetic reagents provide a starting point for the additional functional and ultra-structural studies that will be required to elucidate the circuit mechanisms that produce the response properties of these and other LC cell types.

## LC neuron convergence and the signaling of positional information

The suggestion that LC cells are feature-responsive neurons has been partly based on the apparent dramatic reduction in retinotopy between LC neuron dendrites, which have a retinotopic arrangement in the lobula, and their axons, which appear to discard this spatial information as they converge onto target glomeruli (*Mu et al., 2012*; *Strausfeld and Okamura, 2007*; *Strausfeld et al., 2007*). We extended previous analyses of LC neuron arbor convergence by directly visualizing multiple single LC cells in a glomerulus in the same fly (*Figure 4*). These experiments revealed no detectable retinotopy of LC cell processes in most glomeruli even at this cellular level of resolution. It is possible that the responses of individual LC cells carry information about retinotopic position; given the comparatively small size of LC dendrites (the lateral spread of even the largest LC cells covers less than 20% of visual columns) and the retinotopic distribution of these dendrites in the lobula it would be surprising if they did not. Such retinotopic responses could for example be relevant for those LC cell types that appear to have presynaptic sites in the lobula and are thus likely to provide input to retinotopically organized circuits. However, with the caveat that we did not examine synapse-level connectivity, for most LCs the available anatomical information appears to support the view that much retinotopic information is discarded at the glomerulus level. Consistent with this anatomical observation, the calcium imaging experiments from single LC cell types revealed visual responses to localized stimuli that could be measured throughout a cross-section of the glomerulus without clear retinotopic arrangement of the responding axons (*Figure 12B,C*). Because of the columnar nature and apparently restricted visual field of the dendrites of LC neurons, the features computed by individual LC neurons are likely to be well defined in subregions of the eye, with

perhaps downstream circuits required to integrate these locally-extracted features, as discussed above for looming. We currently have little insight into how these computations are initiated in the optic glomeruli and this remains an exciting area for future investigation.

Unlike the other LC neurons, we found that LC10, and to a lesser extent LC9, cells retain some retinotopic information in the arrangement of their axon terminals indicating that the loss of retinotopy is not a necessary consequence of axonal convergence onto a glomerular target region. More specifically, we observed that the order of LC10 axonal terminals in the AOTu along the DV axis matches the sequence of AP positions of the corresponding dendrites in the lobula. This organization could facilitate synaptic interactions of LC10 cells corresponding to different azimuthal positions in the visual field with distinct target cells. Consistent with a possible general role of the AOTu in the processing or the relaying of retinotopic information, retinotopic responses have recently been observed in the dendrites of central complex neurons (*Seelig and Jayaraman, 2013*) that, mainly based on work in other insects (*Pfeiffer and Homberg, 2014*; *Pfeiffer et al., 2005*), are thought to be synaptic targets of output neurons of the lateral zone of the AOTu (*Ito et al., 2014*).

We found that, independent of the presence or absence of retinotopy at the glomerulus level, positional information can be extracted from the differential activity of LC cells between the two optic lobes. We directly demonstrated this capability by genetically restricting optogenetic LC neuron activation to only one optic lobe. This unilateral activation evoked directional turning responses relative to the activated brain side. Thus, LC neuron signaling appears to convey information on both different visual features and their location. This may further extend the similarities to the antennal lobes where differences in odorant receptor neuron activity between the left and right antennal lobes may contribute to odorant tracking (*Gaudry et al., 2013*).

## Optogenetic activation of LC neurons can induce naturalistic behaviors

We found that activation of different types of LC neurons can induce distinct behaviors including jumping, reaching, wing extension, forward walking, backward walking and turning. While specific activation phenotypes have been reported for a variety of cell types and behaviors, many of these studies have focused on command-like neurons thought to orchestrate specific motor programs (*Bidaye et al., 2014*; *Flood et al., 2013*; *Lima and Miesenböck, 2005*; *von Philipsborn et al., 2011*; *von Reyn et al., 2014*). By contrast, the activation phenotypes we report here result from the optogenetic stimulation of different types of related visual projection neurons. A plausible interpretation of these results is that activation of LC neurons can mimic the presence of the visual features that these neurons normally respond to and thus elicits behavioral responses associated with these fictive stimuli. This possibility is supported by several lines of evidence from our studies of LC6 and LC16. First, optogenetic depolarization of each of these cell types evokes a specific behavioral response—backward walking for LC16 and jumping for LC6—that resembles a similar natural avoidance or escape behavior (*Parigi et al., 2014*; *von Reyn et al., 2014*). Second, backward walking and jumping can both also be elicited by presentation of a predator-mimicking visual loom (*Card and Dickinson, 2008b*) (and this study) and, third, in calcium imaging experiments both LC16 and LC6 showed a preferential response to a similar looming stimulus compared to a number of related stimuli. Although we did not explore LC10 response properties, we note that LC10-activation phenotypes also show similarities to natural behaviors: movements resembling the directed foreleg extension displayed during activation-evoked reaching occur, for example, during gap-climbing behavior (*Pick and Strauss, 2005*) and in aggressive fly-fly interactions (*Chen et al., 2002*).

Overall, the LC neuron activation phenotypes we observed suggest that the encoding of visual information at the level of LC neurons is sufficiently specialized to contribute to distinct behavioral responses in a cell-type dependent fashion. However, patterns of LC neuron activation that produce more refined fictive stimuli than we employed in the current work will be required to fully explore the LC neuron behavioral repertoire. Likewise, more comprehensive physiological studies of the response properties of the LC cell types will be needed.

## Further integration and processing of LC neuron signals by downstream circuits is likely to be required to activate specific behaviors under natural conditions

How does LC cell activation evoke specific behavioral responses? In the simplest scenario, LC neuron depolarization could directly activate a single postsynaptic premotor descending interneuron that would then in turn trigger the observed behavior. This appears plausible in some cases: for example, activation of LC4 neurons (called ColA cells in larger flies) might evoke a jumping response via activation of the Giant Fiber (GF) cells, a pair of large descending neurons known to be postsynaptic to ColA (*Strausfeld and Bassemir, 1983*) and LC4 (K von Reyn and GM Card, personal communication, September 2016) and which have a known role in escape behavior (*von Reyn et al., 2014*; *Wyman et al., 1984*). For other LC cell types, there is currently no evidence suggesting a direct connection to descending neurons. For example, candidate descending neurons for the LC16 backward walking response, the moon-walker descending interneurons (*Bidaye et al., 2014*), do not have dendrites in or near the LC16 glomerulus. Responses to diverse visual stimuli, some of which may derive from LC neuron activity, have also been observed in higher order brain centers without direct connections to LC neurons such as the central complex (*Seelig and Jayaraman, 2013*; *Weir and Dickinson, 2015*).

Our activation experiments also provide several indications that the signaling downstream of LC neurons is likely to be more complex; for example, activation of a single LC cell type can give rise to multiple behaviors such as reaching, wing extension and turning for LC10, or backward walking and turning for LC16. Changes of the spatial pattern of LC neuron activation, as in our stochastic labeling experiments, can further modify activation phenotypes. For example, unilateral LC16 activation primarily evokes turning away from the location of LC16 activation, not backward walking, suggesting that the relative differences in LC16 activity between the two eyes can guide the direction of motor output through downstream signaling. Furthermore, several different LC neuron types may contribute to the same or similar behaviors, as suggested by the jumping phenotypes of LC4, LC6, LC15, LPLC1 and LPLC2. Presumably, visual signals and other information downstream of LC neurons are integrated to select appropriate behavioral actions. Such additional processing is also suggested by the cases of neurons with overlapping response properties but distinct activation phenotypes such as LC6 and LC16. We also note that some responses to LC neuron activation appear to be context dependent; for example, we observed reduced forward walking for several LC cell types on the platform of the single-fly assay that is much smaller than the arena used in the arena assay (*Figure 8— figure supplement 1E and F*, *Supplementary file 1B*).

In addition, we only examined the behavior of standing or walking flies and LC neuron signaling might have different consequences depending on the behavioral state. For example, looming stimuli can also elicit avoidance responses in flying flies (*Muijres et al., 2014*; *Tammero and Dickinson, 2002*), but these responses differ from the takeoff or retreat behaviors of walking animals. Therefore, while LC cell activity appears to convey visual information that is specialized for sets of related behavioral responses, LC neurons do not appear to instruct a single behavioral output.

## Several types of LC neurons may contribute to avoidance and escape circuits

The most common activation phenotypes observed in our screen were apparent avoidance responses. Furthermore, in addition to the LC cells studied here, other VPNs may also contribute to avoidance behaviors (*de Vries and Clandinin, 2012*). This predominance of avoidance phenotypes is perhaps not unexpected. Since escape responses have to be fast and reliably executed under many different conditions, neurons that signal features that can evoke escape may be particularly likely to show phenotypes in an activation screen. Given the importance of predator avoidance for fly survival, it appears plausible that a considerable fraction of visual output neurons might be utilized for the detection of visual threats ranging from looming to small objects (*Card, 2012*; *Maimon et al., 2008*). Furthermore, it is likely that CsChrimson-mediated depolarization of an entire population of LC neurons is more similar to the pattern of neuronal activity induced by an imminent collision, and thus responses of many individual loom-sensitive neurons, so it is not surprising that our activation screen revealed at least two looming-sensitive neuron types.

The escape-inducing neurons we identified could provide inputs to different escape response pathways, such as long- and short-mode escape (*von Reyn et al., 2014*), or act as multiple inputs to the same downstream circuits. Interestingly, neurons with avoidance-like activation phenotypes project to two separate groups of adjacent glomeruli, one in the dorsal PVLP (LC6, LC16 and also LC15) and one more ventral and medial (LC4, LPLC1 and LPLC2). This spatial organization may facilitate synaptic interactions of functionally related LC neuron types with common downstream pathways for a specific behavior. The second group is close to dendritic branches of the GF, large descending neurons required for short-mode responses in *Drosophila* and a postsynaptic partner of LC4/ColA (*Strausfeld and Bassemir, 1983*) (K von Reyn and GM Card, personal communication, September 2016) and possibly also the two LPLC cell types. LC6 terminals do not overlap with GF dendrites and LC6 cells may play a role in the GF-independent escape pathways that have been proposed in both *Drosophila* (*Fotowat et al., 2009*; *von Reyn et al., 2014*) and housefly (*Holmqvist, 1994*). Parallel neuronal pathways involved in escape behaviors have been identified or postulated in both vertebrates and invertebrates (*Burrows and Rowell, 1973*; *Fotowat and Gabbiani, 2011*; *Fotowat et al., 2011*; *Münch et al., 2009*; *Yilmaz and Meister, 2013*), but a contribution of several identified visual projection neurons to such pathways, as suggested by our activation screen, has not been previously reported. Different visual output neurons with distinct tuning of their response properties to looming parameters such as speed, size, luminance change or edge detection might have evolved to ensure robust responses to avoid predators or collisions. It is, however, currently not known whether LPLC1, LPLC2, LC4 and LC15 are indeed sensitive to looming stimuli and if so, whether their response details differ from LC16, LC6 and each other. Nevertheless, the identification of these neurons opens the possibility to examine the potential contribution of several visual pathways to avoidance behaviors.

LC neurons are a subset of the about a hundred VPN cell types that relay the output of optic lobe circuits to targets in the central brain. Our data strongly support existing proposals for LC cell types as feature-detecting neurons, which have been mainly based on the distinct anatomical properties of LC cells (*Strausfeld and Okamura, 2007*). While these anatomical features distinguish LC neurons from many other VPNs, an association of VPN pathways with specific behaviors is not unique to LC cell types. The notion that individual neuronal pathways are tuned for specific behavioral requirements is a prominent theme in invertebrate neuroethology, with these neurons described as 'matched filters' for behaviorally relevant features of the external world (*Warrant, 2016*; *Wehner, 1987*). A number of previously studied VPN pathways, outside of the LC subgroup, have been described as encoding specific behaviorally related visual stimuli. In particular, very similar to our results for LC6 and LC16, a group of tangential cells of the lobula and lobula plate (Foma-1 neurons) were found to respond to looming visual stimuli and, upon optogenetic activation, trigger escape responses (*de Vries and Clandinin, 2012*). And perhaps most famously, the long-studied LPTCs, such as the HS and VS cells, integrate local motion signals so as to preferentially respond to global optic flow patterns that are remarkably similar to visual motion encountered during specific behavioral movements (*Hausen, 1976*, *1982a*; *Krapp et al., 1998*). Both our results and these findings are consistent with the idea that, at the outputs of the fly visual system, we find VPN pathways whose encoding properties are already well matched to particular fly behaviors or groups of behaviors. Matching the response properties of these deep sensory circuits to behavioral needs may be a general evolutionary solution to the challenge of dealing with the complexity of the visual world with limited resources.

## Concluding remarks

LC neurons have long been recognized as a potential entry point for the circuit-level study of visual responses outside of the canonical motion detection pathways. We provide a comprehensive anatomical description of LC cell types and genetic reagents to facilitate such further investigations. We also show that activation of several LC cell types results in avoidance behaviors and that some of these same LC types respond to stimuli that can elicit such behaviors. Other LC neurons appear to mediate attractive behavioral responses. Our work provides a starting point for exploring the circuit mechanisms both upstream and downstream of LC neurons.

## Materials and methods

### Fly stocks and rearing conditions

Split-GAL4 transgenes were selected based on GAL4-line expression patterns (*Jenett et al., 2012*; *Kvon et al., 2014*) (Barry J Dickson, personal communication) and constructed as previously described (*Pfeiffer et al., 2010*). Tables with details of genotypes are included as *Supplementary file 1B, 1C and 1D*. *Supplementary file 1B* summarizes genotypes and results of behavioral experiments. *Supplementary file 1C* lists all LC cell types described here and the driver lines for each type; *Supplementary file 1D* provides details of the genetic reagents used for each panel of the anatomy Figures and for *Videos 1* and *2*. With a few exceptions indicated in *Supplementary file 1B and 1C*, all split-GAL4 AD transgenes are inserted in *attP40* and all DBD transgenes in *attP2*.

*Drosophila melanogaster* flies were reared on standard cornmeal/molasses food at 25°C and 50% humidity unless otherwise indicated. For optogenetic activation experiments in the arena and stochastic activation experiment, flies were reared on standard food supplemented with retinal (0.2 mM all-trans-retinal prior to eclosion and then 0.4 mM post eclosion) at 22°C and 60% humidity in darkness. For optogenetic activation experiments in the single-fly assay, flies were reared on standard food supplemented with retinal (0.4 mM all-trans-retinal throughout) at 22°C and 60% humidity in darkness. Flies of both sexes were used for behavioral experiments unless otherwise indicated. All anatomical analyses were done with female flies.

Vitamin A-deficient food was described previously (*Nichols and Pak, 1985*). Briefly, per 500 ml of food: 270 ml $H_2O$, 230 ml grape juice (Welch's, Welch Foods Inc., Concord, MA), 11 g Bacto-agar (Difco, Franklin Lakes, NJ), 30 g glucose, 10 g sucrose, 5 g fructose, 10 g yeast (Fleischmann's dry), 10 ml 1 M NaOH, 2 ml proprionic acid and 0.2 ml phosphoric acid.

In addition to the split-GAL4 lines (see *Supplementary file 1B, 1C and 1D*), the following fly strains were used: (1) 20XUAS-CsChrimson-mVenus in *attP18* (*Klapoetke et al., 2014*); (2) pBDPGAL4U in *attP2*, an enhancerless GAL4 driver (*Pfeiffer et al., 2010*); used as a control driver in behavioral assays; (3) pBPhsFLP::2PEST in *attP3* (*Nern et al., 2015*); (4) pJFRC300-20XUAS-FRT>-dSTOP-FRT>-CsChrimson-mVenus in *attP18*; created by transferring the stop cassette from pJFRC177-10XUAS-FRT>-dSTOP-FRT>-myr::GFP (*Nern et al., 2011*) into UAS-CsChrimson-mVenus; (5) MCFO-1 (*Nern et al., 2015*); (6) MCFO-7 (*Nern et al., 2015*); (7) pJFRC200-10XUAS-IVS-myr::smGFP-HA in *attP18*, pJFRC216-13XLexAop2-IVS-myr::smGFP-V5 in *su(Hw)attP8* (HA_V5) (*Nern et al., 2015*); (8) pJFRC7-20XUAS-IVS-GCaMP6m in *VK00005* (*Chen et al., 2013*) in DL background; (9) w;; pJFRC51-3XUAS-IVS-syt::smHA in *su(Hw)attP1*,pJFRC225-5XUAS-IVS-myr::smFLAG in *VK00005* (*Nern et al., 2015*); (10) *norpA[36]* (*de Vries and Clandinin, 2012*); (11) w+ DL; DL; pJFRC49-10XUAS-IVS-eGFPKir2.1 in *attP2* (DL) (DL denotes chromosomes derived from the DL fly strain)(*von Reyn et al., 2014*); (12) DL (wild-type strain from M.H. Dickinson, University of Washington).

### Immunohistochemistry and imaging

For screening of split-GAL4 combinations, we adopted a protocol previously applied to large-scale characterization of GAL4 expression patterns of larval fly brains (*Li et al., 2014*). Brains of adult flies carrying the split-GAL4 hemidriver combination of interest and a UAS reporter (pJFRC200-10XUAS-IVS-myr::smGFP-HA in *attP18*) were dissected in insect cell culture medium, incubated ~12–24 hr at 4°C in 10 μl 1% (w/v) paraformaldehyde in the same medium in 60-well or 72-well Terasaki plates, washed with PBS and subsequently attached to Poly-L-Lysine coated coverslips while immersed in PBS. Further processing was in Copeland jars in a total volume of ~10 ml. This included the following steps: 2 × 5 min in PBT (PBS + 0.5% Triton X-100), 1 hr in PBT with 0.5% Normal Goat Serum (PBT-NGS), overnight at 4°C in PBT-NGS with primary antibodies (rabbit anti-GFP 1:1000, mouse anti-Brp or mAb Nc82 (*Wagh et al., 2006*) 1:50), 2 × 5 min in PBT, 1 hr in PBT-NGS, overnight at 4°C in PBT-NGS with secondary antibodies, 2 × 5 min in PBT and 1 × 5 min in PBS. Brains were then post-fixed for 4 hr with 4% (w/v) PFA in PBS, further rinsed with PBS and subsequently dehydrated and mounted in DPX as described (*Nern et al., 2015*). A detailed updated version of this screening protocol is available online (https://www.janelia.org/project-team/flylight/protocols under 'IHC - Adult Split Screen').

We used two sets of markers to visualize split-GAL4 expression patterns. 20XUAS-CsChrimson-mVenus in *attP18* (*Klapoetke et al., 2014*) was used to reveal the overall expression patterns of split-GAL4 lines used in behavioral experiments with this effector. For most other anatomical analyses (except stochastic labeling), a combination of pJFRC51-3XUAS-IVS-Syt::smHA in *su(Hw)attP1* and pJFRC225-5XUAS-IVS-myr::smFLAG in *VK00005* (*Nern et al., 2015*) was used. Immunolabeling of fly brains to detect these markers together with anti-Brp as a neuropil label was performed as described (*Aso et al., 2014a*). Brains were mounted in DPX. Detailed protocols can be found online (https://www.janelia.org/project-team/flylight/protocols under 'IHC - Anti-GFP', 'IHC - Polarity Sequential' and 'DPX mounting').

Stochastic labeling of LC neurons in multiple colors was performed using Multicolor FlpOut (MCFO) (*Nern et al., 2015*). MCFO fly stocks used for specific experiments are listed in *Supplementary file 1D*. MCFO samples were processed for immunolabeling of three epitope-tagged marker proteins (smGFP-HA, smGFP-V5 and smGFP-FLAG [*Viswanathan et al., 2015*]) together with the anti-Brp reference pattern and mounted in DPX as described (*Nern et al., 2015*). Detailed protocols can be found online (https://www.janelia.org/project-team/flylight/protocols under 'IHC - MCFO').

Images were acquired on Zeiss LSM 710 or 780 confocal microscopes with 20 × 0.8 NA or 63 × 1.4 NA objectives at 0.62 µm x 0.62 µm x 1 µm (20x) or 0.19 µm x 0.19 µm (in a few cases 0.38 µm x 0.38 µm) x 0.38 µm (63x) voxel size. In some images (e.g. panel A of *Figure 3—figure supplement 2*), signal from AlexaFluor 647 or DyLight 649 dyes was also detected in the reference pattern (Alexa488) channel. This crosstalk appears to be mainly due to altered spectral properties of these dyes in the DPX mounting medium rather than microscope set-up or antibody cross-reaction. Four channel MCFO images (HA, V5, FLAG plus anti-Brp) were acquired as two separate stacks which were combined post-imaging. For 63x images, brain regions larger than a single field of view were imaged as up to five overlapping tiles; multiple tiles were combined ('stitched'). Brain alignment to a template brain was achieved as described (*Aso et al., 2014a*); to facilitate alignment of 63x tiles, most samples imaged as 63x were also imaged as whole brains at 20x. Initial image processing steps applied to most or all images (such as stitching and alignment) were as previously described (*Aso et al., 2014a*). Alignment quality showed some variation between specimens and within subregions of the same specimen; samples with acceptable alignment quality in the relevant brain regions were identified by visual inspection of an overlay of the aligned anti-Brp patterns of the sample and the template brains. Some processed (e.g. aligned or stitched) images used for anatomical analyses were stored using a 'visually lossless' compression (h5j format). This compression did not appear to have a detectable effect on the neuroanatomical features that were characterized using these images.

Fiji (http://fiji.sc) and Vaa3D (*Peng et al., 2010*) were used for most analyses and processing of individual images. To generate specific views from three dimensional image stacks, appropriately oriented substack views were generated using the Neuronannotator mode of Vaa3D and exported as TIFF format screenshots. The scale of these images was calibrated using the known dimensions and pixel resolution of the starting image and the pixel resolution and zoom of the exported image. In some cases, multiple figure panels (in either the same figure or different figures) show different views of the same cells that were generated from the same image stack (using Vaa3D) to illustrate distinct anatomical features. To display images in similar orientations within a figure, some images were rotated or mirrored. To fill in empty space outside the original field of view in some panels with rotated images, canvas size was increased and space outside the original image filled in with zero pixels using Fiji. Some images (for example in the overlays in *Figure 3*) were manually segmented to remove background or labeled cells or structures other than those of interest; instances of such processing are specifically noted in the respective figure legends. Overlays of aligned images were assembled in Fiji with the exception of *Figure 11A* for which aligned images of LC6 and LC16 were segmented and overlaid using FluoRender (*Wan et al., 2012*), as previously described (*Aso et al., 2014a*). The figures were assembled using Adobe Indesign with some schematics generated in Adobe Illustrator.

All reported anatomical features were confirmed with multiple specimens. For the analyses of glomerulus shape and location and overall lobula layer patterns, at least three brains per cell type were imaged at high resolution (63x) with pJFRC51-3XUAS-IVS-Syt::smHA in *su(Hw)attP1* and pJFRC225-5XUAS-IVS-myr::smFLAG in *VK00005* as reporters (see above). For the illustrations in *Figures 3* and

**5**, individual aligned brains were selected based on alignment quality in the regions of interest; however the features shown were also examined in unaligned samples and for at least two additional brains. The number of MCFO labeled single cells that were imaged and examined at high resolution varied: the lowest numbers was for LC26 (7 cells); the highest numbers were for LC10 subtypes (see *Figure 10—figure supplement 2B*). Estimates of cell numbers in *Supplementary file 1A* are based on manual counts using high resolution (63x) confocal stacks of the indicated split-GAL4 driver lines with pJFRC225-5XUAS-IVS-myr::smFLAG in *VK00005* as reporter (see above). Numbers are averages of counts of cells from three or more optic lobes. Lateral arbor spreads of LC neurons within lobula layers were estimated using substack projection images similar to those shown in *Figure 7—figure supplement 1*. A segmented line was drawn along the part of a lobula layer covered by a cell's arbors and the length of this line (measured using Fiji) divided by the maximum length of the entire layer determined in the same way. Arbor spreads were measured along both the AP and DV axes of the lobula. To estimate visual column coverage from these numbers, we assumed a circular eye of ~750 ommatidia, a uniform distribution of the corresponding visual columns across the lobula (i. e. ~31 columns along each lobula axis) and treated LC neuron arbors within each layer as planar and ellipse-shaped. All of these simplifications, which are approximations of described features of the visual system (*Wolff and Ready, 1993*; *Meinertzhagen and Hanson, 1993*), limit the precision of these estimates, with larger overestimates expected for large arbors.

## Optogenetic activation behavioral assays

### Circular arena assay

Groups of approximately 25 flies (3 to 10 d post-eclosion) were tested at 25°C and 50% relative humidity in a dark chamber. Optogenetic activation experiments were performed in a 100 mm diameter and 3 mm high circular arena as previously described (*Aso et al., 2014b*; *Klapoetke et al., 2014*). For the activation of neurons expressing CsChrimson, the arena was uniformly illuminated with 617 nm LEDs (Red-Orange LUXEON Rebel LED - 122 lm; Luxeon Star LEDs, Brantford, Canada) at increasing intensities: 5, 10, 20, 49 and 94 $\mu W/mm^2$. Overall, higher light intensities appeared to produce more penetrant but otherwise qualitatively similar behavioral responses. Data collected with the maximum light stimulus (94 $\mu W/mm^2$) were therefore used for all detailed analyses. For each intensity, five to six 1 s trials were performed. The inter-stimulus interval was 9 s between trials of the same intensity and 20 s between trials of different intensities. Videos were recorded under reflected IR light using a camera (ROHS 1.3 MP B and W Flea3 USB 3.0 Camera; Point Grey, Richmond, Canada) with an 800 nm long pass filter (B and W filter; Schneider Optics, Hauppauge, NY) at 50 frames per second, 1024 × 1024 pixel resolution.

### Single-fly assay

Flies were automatically released one at a time onto a small glass platform (5 mm by 5 mm) using a custom-built system. Further details of this system will be described elsewhere. Two small prisms, one placed in front of the glass platform and one below it, allowed a side view and a bottom view of the fly to be simultaneously recorded on a single high-speed video camera (SA-4 and SA-X, Photron, San Diego, CA). A 10 Hz video feed was processed in real time and used to coordinate the stimulus presentation via software (https://github.com/wryanw/single_fly_tracking_and_analysis) written in MATLAB (Mathworks, inc. Natick, MA). For optogenetic activation, four 608 nm LEDs were turned on for 50 ms with 3 mW total intensity beginning when the fly was still and centered on the platform. For looming stimulus experiments, a dark disc on a light background was projected on a screen above the platform with a non-linear expansion rate that mimicked that of an object approaching the fly with constant velocity (see *von Reyn et al. [2014]*). Video frames of the fly behavior were recorded at a rate of 6000 frames per second and a spatial resolution of 3 pixels per mm for the duration of each stimulus plus 500 ms after for optogenetic experiments and 150 ms after for looming experiments. For the looming-evoked jumping experiments, the looming stimulus was presented at an azimuth of 90°, which elicits jumping more frequently than any other azimuth (data not shown). For backward walking, the stimulus was presented at 0°, which elicits that behavior better than the 90° azimuth stimulus used to elicit jumping. Both stimuli were presented at an elevation of 45° above the horizon. All experiments were conducted at 23°C and 50% humidity. Data were acquired in 20 min sessions on four independent apparatuses simultaneously, during which time about 20 videos

could be collected per apparatus. Videos were used in subsequent analyses only when one fly was present for the duration of the experiment (90% of videos).

## Stochastic labeling and activation assay

Crosses of 40–50 pairs of males and females were set up in egg-laying cages on grape juice plates (containing 30% agar) supplemented with a drop of yeast paste. First instar larvae hatched within a 3 hr period were collected with a brush, seeded in standard cornmeal/molasses food supplemented with retinal and were immediately subject to heat shock at 37°C for 90–120 min to induce stochastic, recombinase-mediated excision of transcriptional-terminating cassettes in the 20XUAS-FRT>STOP>FRT-CsChrimson-mVenus in *attP18* transgene. Flies were then reared at 22°C at 60% relative humidity in darkness. The activation behaviors of individual 4–10 d post-eclosion flies were tested at 25°C at 50% relative humidity in a 32 mm diameter bowl arena with a top dome (6 mm maximum height) that had been coated with Sigmacote (Sigma-Aldrich Corp., St. Louis, MO).

Bowl arenas were illuminated with 617 nm LEDs at 94 $\mu$ W/mm$^2$ and fifteen 1 s trials with an inter-stimulus interval of 9 s were performed for each fly. Trials in which flies showed startle response or stayed on the ceiling (two trials per fly on average) were excluded from further analysis. Video recording and fly tracking were performed in the same way as described for the circular arena assay. After assaying, flies were retrieved and their brains were dissected and stained to assess CsChrimson-mVenus expression. Behavioral data were categorized by expression patterns (bilateral, unilateral on the right, unilateral on the left, or no labeling). In the LC10 experiments, some of the assayed flies showed bilateral CsChrimson expression but with more labeling on one side compared to the other. These flies were placed in their own category and behavioral data for this group analyzed separately. For unilateral labeling, trajectories for left activation were inverted and combined with right activation data. Flies of both sexes were used in the LC16 experiments and only female flies were used in the LC10 experiments.

## Behavioral data analysis and statistics

Fly tracking videos were analyzed using either manual human annotation or software (https://github.com/wryanw/bowl_assay_tracking_and_analysis) written in MATLAB. The automated tracking identifies the center of mass of each fly and determines the fly's azimuthal orientation for every frame using a template-matching algorithm. The center of mass is determined in two dimensions in the arena assay and three dimensions in the single-fly assay, with 95% of the data having an error of less than five pixels. Position and orientation information for each fly was converted to velocity (to measure forward and backward walking) using a smoothing filter or angular velocity (to measure turning) using a sliding Savitzky-Golay filter (*Orfanidis, 1996*). Forward and backward walking velocity was determined relative to the orientation of the fly, along its path of movement. For the arena assay, mean velocities were calculated for the full 1 s of optogenetic activation. For the single-fly assay, mean velocities for motion and turning were calculated for the first 350 ms after the onset of light stimulation during which most flies remained in the field of view on the assay's small platform. Reaching and jumping were elicited later than 350 ms in some cases, so a 500 ms cutoff was used for these behaviors. These average motion and turning values for individual flies were used to generate the distributions seen in *Figure 8—figure supplement 1* and *Figure 9—figure supplement 2*. For assessing behavioral phenotype penetrance, we used thresholds that required turning rates for experimental data to be beyond the 97.7th percentile of control data and walking velocity to be outside of the 2.3rd percentile (negative values, backward walking) or the 97.7th percentile (positive values, forward walking) to be scored as positive. These percentiles roughly correspond to two standard deviations away from the mean in a normal distribution; we use percentiles because the distributions are non-normal. The thresholds were determined independently for the arena and the single-fly assays and are indicated by horizontal red lines in *Figure 8—figure supplement 1* and *Figure 9—figure supplement 2*. The thresholds were used to convert mean velocities into true/false values for forward walking, backward walking and turning. Jumping and reaching were scored manually. Jumping in the arena assay was defined as events in which a fly's legs all left the ground, usually accompanied by the initiation of wing flapping and change in fly shape. Jumping in the single-fly assay was defined as a fly leaving the platform due to a sudden leg extension. Reaching in the arena assay was defined as when the fly pitches its body back and reaches with a forward and upward

extension of the forelegs, while staying in the same location. Reaching in the single-fly assay was defined as a head-up elevation of the long body axis, with both front legs leaving the platform and at least one leg elevating above a horizontal plane at the level of the dorsal tip of the head. For box plots, the dividing line in the box indicates the median, the boxes contain the inner quartile range, the notches give the 95% confidence interval, the lines extending beyond the box include 95% of the data, and the dots beyond those lines are outliers.

## Ca$^{2+}$ imaging, data analysis, and visual stimuli

All flies used for calcium imaging experiments were reared under standard conditions (25°C, 60% humidity, 12 hr light/12 hr dark, standard cornmeal/molasses food), and all imaging experiments were performed on females 3–6 d post-eclosion. To image from individual lobula columnar cell-types, split-GAL4 driver lines (LC6: OL0070B, LC16: 0L0046B, LC11: OL0015B) were crossed to pJFRC7-20XUAS-IVS-GCaMP6m in *VK00005* (DL background) effector line. The imaging preparation was almost identical to that described in (*Strother et al., 2014*). Briefly, flies were cold anesthetized and tethered to a fine wire at the thorax using UV-curing adhesive. The two most anterior legs (T1) were severed and glued down along with the proboscis to prevent grooming of the eyes and to immobilize the head. Tethered flies were glued by the head capsule into the fly holder and after addition of saline to the bath, the cuticle at the back of the head was dissected away to expose the brain. Muscles 1 and 16 were severed to reduce the motion of the brain within the head capsule, and the post-ocular air sac on the imaged side was removed to expose the optic glomeruli.

The optic glomeruli were imaged using a two-photon microscope (Prairie Ultima IV, Bruker Optics Inc., Billerica, MA) with near-infrared excitation (930 nm, Coherent Chameleon Ultra II, Coherent Inc., Santa Clara, CA) and a 60x objective (Nikon CFI APO 60XW). The excitation power was never greater than 20 mW at the sample. Imaging parameters varied slightly between experiments but were within a small range of our typical acquisition parameters: $128 \times 90$ pixel resolution, and 10 Hz frame rate (10.0–10.5 Hz). LC cell axon calcium data were collected from single planes selected to capture a consistently large slice of each glomerulus.

Flies were placed in the center of a modular LED display (*Reiser and Dickinson, 2008*) on which visual stimuli were presented. The display was configured to cover 60% of a cylinder, with LEDs subtending 72° in elevation and 216° in azimuth (maximum pixel size of 2.25°) as seen by the fly in the center of the cylinder. The display consists of 574 nm peak output LEDs (Betlux ultra-green $8 \times 8$ LED matrices, #BL-M12A881UG-XX, Betlux, Ningbo, China) covered with a gel filter (LEE #135 Deep Golden Amber) to greatly reduce stimulus emission at wavelengths that overlap with those of GCaMP emission.

The stimuli were generated using custom MATLAB scripts (https://github.com/mmmorimoto/visual-stimuli). The dark loom stimulus consisted of a series of 35 disk sizes, with the edge pixel intensity interpolated to approximate a circle on the LED screen. The luminance-matched stimulus was created using the dark looming disk stimulus, spatially scrambling the location of dark pixels of each frame only within the area of the final size of the disk. The time series of looming stimuli sizes were presented based on the classic parameterization for looming stimuli assuming a constant velocity of approach. The speed of the loom is represented by a single parameter (r/v) that describes the ratio of the stimulus radius to its approach speed (*Gabbiani et al., 1999*). All looming stimuli appear as 4.5° spots and increase to a maximum diameter of 54°. The experimental protocol consisted of 3 repetitions of each stimulus type presented using a randomized block trial structure.

Data analysis was performed with software written in MATLAB. Motion compensation was performed by cross-correlating each frame to a reference image, using software written by James Strother (https://bitbucket.org/jastrother/neuron_image_analysis). The fluorescence signal is determined within hand-drawn regions of interest selected to tightly enclose the entire slice of each glomerulus captured within the imaging plane. $\Delta F/F$ is calculated as the ratio of $(F - F_0) / F_0$, where F is the instantaneous fluorescence signal and $F_0$ is calculated as the 10th percentile of the fluorescence signal within a sliding 300 frame window. For combining the responses of individual flies across animals, we normalized the $\Delta F/F$ responses from each individual fly to the 98th percentile of the $\Delta F/F$ across all visual stimuli within one experiment. All responses are the mean of the mean response (across repeated stimulus presentations) of each of five flies. Error bars indicate mean ± SEM. All significance results presented for Ca$^{2+}$ imaging were determined with the Mann-Whitney test.

## Acknowledgements

We thank the Janelia FlyLight Project Team and Teri Ngo for help with brain dissections, histology and confocal imaging and Janelia Scientific Computing for image processing and data analysis tools. Heather Dionne made the pJFRC300 construct. Yoshi Aso shared an early version of the arena assay and Steve Sawtelle, Igor Negrashov and Jinyang Liu made improvements to this apparatus. Ji-Young Kim provided the arena used for the stochastic activation experiments. We also thank William Rowell for technical assistance, James Strother for the Ca²⁺ imaging software, Tom Clandinin for the *norpA* flies, Zhiyong Liu, Rajyashree Sen, Yoshi Aso and Wyatt Korff for advice, Kei Ito, Alice Robie, Kristin Branson, Ines Ribeiro and Barry Dickson for communicating unpublished results, Axel Borst for comments on the manuscript, Andrew Straw and colleagues for agreeing to a shared nomenclature for the LC cell types and Crystal Di Pietro for administrative assistance. Ming Wu was a Helen Hay Whitney Foundation Postdoctoral Fellow.

## Additional information

### Funding

| Funder | Author |
| --- | --- |
| Howard Hughes Medical Institute | Ming Wu<br>Aljoscha Nern<br>W Ryan Williamson<br>Mai M Morimoto<br>Michael B Reiser<br>Gwyneth M Card<br>Gerald M Rubin |

The funders had no role in study design, data collection and interpretation, or the decision to submit the work for publication.

### Author contributions

MW, Conceived the study, Generated the split-GAL4 lines and developed the unilateral activation approach, Performed the behavioral experiments and analyses, Contributed to data analysis and writing of the paper; AN, Conceived the study, Performed the anatomical characterization of the LC cells, Generated the split-GAL4 lines and developed the unilateral activation approach, Contributed to data analysis and writing of the paper; WRW, Performed the behavioral experiments and analyses, Contributed to data analysis and writing of the paper; MMM, Performed the calcium imaging experiments, Contributed to data analysis and writing of the paper; MBR, Provided input on the calcium imaging experiments, Contributed to data analysis and writing of the paper; GMC, Provided input on the behavioral experiments and analyses, Contributed to data analysis and writing of the paper; GMR, Conceived the study, Contributed to data analysis and writing of the paper

### Author ORCIDs

Ming Wu, http://orcid.org/0000-0002-2193-8271
Aljoscha Nern, http://orcid.org/0000-0002-3822-489X
Michael B Reiser, http://orcid.org/0000-0002-4108-4517
Gerald M Rubin, http://orcid.org/0000-0001-8762-8703

## Additional files

### Supplementary files

• Supplementary file 1. Four tables with information on LC neuron anatomy, results of behavioral experiments and experimental genotypes. (A) Summary of the anatomical properties of the 22 LC neuron types described in detail in this study. Cell numbers and lateral arbor spreads are listed as mean ± SD. Arbors sizes in visual column units were estimated from the measurements of lateral arbor spread. Further details are provided in the Materials and methods. (B) Details of the behavioral experiments in *Figures 8–13*. This table includes information on split-GAL4 hemidrivers (the AD and DBD halves), the behavioral penetrance for each of the five examined phenotypes, and trial and fly

counts, from both the arena and single-fly assays. While the raw data counts within each assay are the same, a small number of trials could not be scored by either manual annotation or automatic tracking; as a result, there are some small differences in the number of quantified data points for the two scoring methods. Use of fly culture media different from standard cornmeal molasses food with supplemental retinal is indicated as follows: ret-: standard cornmeal molasses food without supplemental retinal. Vit-: Vitamin A-deficient food based on the grape juice recipe (see Materials and methods) with supplemental retinal. Vit-, ret-: Vitamin A-deficient food without supplemental retinal. norpA indicates flies that are rendered blind by a null mutation in the norpA gene, *norpA[36]*. (C) Split-GAL4 driver lines for the LC neuron types described in this study. Split-GAL4 line names in bold indicate drivers used in behavioral experiments. (D) Details of the fly lines used to generate the data for the anatomy Figures and *Videos 1* and *2*.

---

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
