## [Decision Letter]

Thank you for submitting your article "Visual projection neurons in the *Drosophila* lobula link feature detection to distinct behavioral programs" for consideration by *eLife*. Your article has been reviewed by three peer reviewers, and the evaluation has been overseen by a Reviewing Editor and K VijayRaghavan as the Senior Editor. The reviewers have opted to remain anonymous.

The reviewers have discussed the reviews with one another and the Reviewing Editor has drafted this decision to help you prepare a revised submission. We hope you will be able to submit the revised version within two months.

Summary:

This manuscript details a fascinating and important part of the visual pathway in flies, visual projection neurons that supply "optic glomeruli" within the ventrolateral protocerebrum. The authors proceed with a three-pronged approach to analyze the neuroanatomy, physiology, and behavioral function of one class of VPNs, the lobula columnar neurons (LCs). They perform an exhaustive and outstanding anatomical analysis and generate sparse genetic drivers (split Gal4) that will be an invaluable resource for the research community. The behavioral screen of all the lines and the calcium imaging of a select subset provide useful information about the function of these neurons.

Taken together, the reagents and data presented in this work represent the most important single contribution ever made to our understanding of *Drosophila* VPNs. The technical quality of the data is uniformly high, and the authors have gone to great lengths to perform relevant controls and useful checks. We therefore strongly support publication of this study in *eLife*.

Essential revisions:

1) A table describing the anatomy of each LC would complement the images and serve as a great reference manual for the community. (e.g. table with columns for dendritic arbors, arbor size, soma position, axonal tracks and presynaptic terminals.) Dendritic tiling is another useful indicator of cell type that should be included.

2) The authors detail the anatomy of LPLC2 and hypothesize that it detects looming stimuli. The activation behavior is consistent with this notion. The authors test GCaMP responses to a looming stimulus in other LCs and should test LPLC2. Otherwise, they should leave out the detailed anatomical evidence until they perform functional studies.

3) There are two reports that came out recently on the neuroanatomy of optic glomeruli, one of which is absent from the citation list and the other of which is under-cited. The Introduction should briefly discuss prior advances by Panser et al. and Costa et al. on the neuroanatomy of VPNs and exactly what gaps are filled by this new work.

4) The authors miss an opportunity to put their work on visual output pathways into a slightly broader context in the fly field. The idea that visual output neurons of various types might be selective for particular visual features, and coupled to specific subsets of the behavioral repertoire is an idea that has now come up at least three times (matched filters of LPTCs in blowfly, Foma-1 neurons having loom selective responses and mediating escape, previous anatomical and physiological characterization of LC neurons). Now that the authors have looked at a large number of new outputs, it would seem like this idea has at least broad support that could be usefully connected.

5) Figure 11: The authors should report the effect of LC6 and LC16 activation on both takeoffs and backward walking. This is an important issue because it pertains to the general conceptual point that visual glomeruli may (or may not) map onto specific individual behaviors.

6) The authors demonstrate that optogenetic activation of two of the LCs strongly trigger behavioral responses that closely match those generated by a looming visual stimulus. Thus, these two LCs are sufficient to drive these behaviors. But are these LCs also necessary for looming evoked escape?

---

## [Author Response]

*Essential revisions:*

*1) A table describing the anatomy of each LC would complement the images and serve as a great reference manual for the community. (e.g. table with columns for dendritic arbors, arbor size, soma position, axonal tracks and presynaptic terminals.) Dendritic tiling is another useful indicator of cell type that should be included.*

We have included a table ([Supplementary-material SD1-data]) summarizing key properties of LC neuron cell types. This table contains information on approximate cell numbers, approximate lateral arbor spreads, layer patterns, presynaptic sites and general arbor structures in the lobula, approximate cell body positions and selected additional features (e.g. retinotopy of LC10s in AOTu, LPLC dendrites in the LP).

As we note in the Results, we did not observe strict dendritic tiling in the lobula for any of the LC types (with the possible exception of LC10a); estimates from cell body counts and approximate arbor spread as well as direct co-labeling of overlapping cells in MCFO experiments indicate that the coverage of one or more lobula layers is > 1 for all or nearly LC types. Overlap between processes of cells of the same type is also common in the medulla; our previous study [Nern, Pfeiffer and Rubin, PNAS 2015] of Dm neurons using similar methods provides detailed examples of both overlapping and strict tiling patterns of medulla neurons.

We added to the Results:

“Taken together, our estimates of the number of LC cells of each type and their approximate dendritic arbor spread within lobula layers suggest that the dendrites of a given LC type, with the possible exception of the small arbors of LC10a, do not show a strict tiling pattern but rather overlap ([Supplementary-material SD1-data]). […] Overlap between processes of cells of the same type is also common in the medulla; our previous study (Nern et al., 2015) of Dm neurons using similar methods provides detailed examples of both overlapping and strict tiling patterns of medulla neurons.”

We also note that all of the 22 LC types described in detail appear to cover the entire array of visual column (though the driver line(s) for some cell types may have slightly incomplete expression pattern and we did not have subtype specific drivers for LC10b and LC10c). Approximately uniform coverage of the lobula array was indeed one of the criteria used for including a cell type in this work: “We searched for cell types that consisted of many similar cells which as populations covered the entire array of visual columns in the lobula and whose axonal projections converged onto single glomerulus-like regions in the ipsilateral central brain (Figure 1).”

*2) The authors detail the anatomy of LPLC2 and hypothesize that it detects looming stimuli. The activation behavior is consistent with this notion. The authors test GCaMP responses to a looming stimulus in other LCs and should test LPLC2. Otherwise, they should leave out the detailed anatomical evidence until they perform functional studies.*

While additional functional studies on LPLC2 are currently in progress, we consider it impractical to include these data in the current manuscript. We therefore (reluctantly) follow the reviewers’ suggestion and have removed the detailed discussion of LPLC2 anatomy and its potential implications (with the plan to report these in the next few months together with the functional work). We deleted some LPLC2 related text, replaced the main Figure 7 with the former Figure 7—figure supplement 1 and added a new supplemental Figure (now Figure 7—figure supplement 2) with more concise information on LPLC arbors in lobula and lobula plate.

*3) There are two reports that came out recently on the neuroanatomy of optic glomeruli, one of which is absent from the citation list and the other of which is under-cited. The Introduction should briefly discuss prior advances by Panser et al. and Costa et al. on the neuroanatomy of VPNs and exactly what gaps are filled by this new work.*

We have added citations to these two papers, acknowledging that they have extended the prior analyses of Otsuna and Ito using computational analyses of images collected by others (combined with some follow-up experiments in the case of Panser et al). We specifically credit Panser et al. with independently proposing a glomerular map, which largely concurs with our more detailed map; we found two additional LC cell types and provide higher resolution imaging that resolves glomeruli that were partly merged in Panser et al. In addition to shape and position of the target glomerulus, we also characterize many other anatomical features (such as lateral spread and layer patterns of lobula arbors, cell body positions and cell numbers) of LC neurons, significantly extending the analyses in the other studies.

We expanded the Introduction as follows:

“In parallel to our work, two recently published independent studies (Costa et al., 2016; Panser et al., 2016) have made use of existing image data (such as (Chiang et al., 2011; Jenett et al., 2012)) to reveal additional VPN pathways from the lobula to optic glomeruli. […] Potential retinotopy within optic glomeruli has not been examined in detail and images with sparse labeling of LC neurons have been interpreted as arguing either for or against such axonal retinotopy (Otsuna and Ito, 2006; Panser et al., 2016).”

We added the following in the Results:

“Costa et al. (Costa et al., 2016) also propose several new LC10 subtypes based on computational clustering of single cell data. […] In agreement with this possibility, attempts to identify GAL4 lines markers for such regionally restricted LC10 subtypes were largely unsuccessful (Panser et al., 2016). By contrast, our split-Gal4 driver lines provide genetic support for the LC10 subdivisions we describe here.”

However, the anatomical analyses reported in these papers are much more limited than what we have included, making precise comparisons difficult if not impossible. Since these papers do not claim to provide such detailed analyses, we feel it would be unfair (and impolite) to criticize them for these shortcomings more than we already have done in the added text quoted above as would be required if we were to describe “exactly what gaps are filled by this new work”.

*4) The authors miss an opportunity to put their work on visual output pathways into a slightly broader context in the fly field. The idea that visual output neurons of various types might be selective for particular visual features, and coupled to specific subsets of the behavioral repertoire is an idea that has now come up at least three times (matched filters of LPTCs in blowfly, Foma-1 neurons having loom selective responses and mediating escape, previous anatomical and physiological characterization of LC neurons). Now that the authors have looked at a large number of new outputs, it would seem like this idea has at least broad support that could be usefully connected.*

We thank the reviewers for this suggestion and have added the following paragraph to the Discussion:

“LC neurons are a subset of the about a hundred VPN cell types that relay the output of optic lobe circuits to targets in the central brain. […] Matching the response properties of these deep sensory circuits to behavioral needs may be a general evolutionary solution to the challenge of dealing with the complexity of the visual world with limited resources.”

*5) Figure 11: The authors should report the effect of LC6 and LC16 activation on both takeoffs and backward walking. This is an important issue because it pertains to the general conceptual point that visual glomeruli may (or may not) map onto specific individual behaviors.*

In Figure 11 we described the LC6 activation takeoff behavior and the LC16 activation backward walking behavior providing in-depth characterization of their highly penetrant optogenetic activation phenotypes first observed in our activation screen reported in Figure 8. Although there is a small percentage of backward walking (23% in the arena assay and 14% in the single-fly assay) for LC6 and jumping (9% in the arena assay and 18% in the single-fly assay) for LC16 ([Supplementary-material SD1-data]), these penetrances are under the 25% threshold we set to be scored as having a phenotype (Figure 8). Therefore, we did not further characterize LC6 backward walking or LC16 jumping phenotype in Figure 11. The general conceptual point that visual glomeruli may not map onto specific individual behaviors is discussed in the Discussion section under the subheading “Further integration and processing of LC neuron signals by downstream circuits is likely to be required to activate specific behaviors under natural conditions”. Based on multiple behavioral data, we concluded in the original paper “while LC cell activity appears to convey visual information that is specialized for sets of related behavioral responses, LC neurons do not appear to instruct a single behavioral output”.

We added the following sentence in the Results:

“The table in [Supplementary-material SD1-data] includes some additional behavioral responses that were below the 25% penetrance threshold used in Figure 8.”

*6) The authors demonstrate that optogenetic activation of two of the LCs strongly trigger behavioral responses that closely match those generated by a looming visual stimulus. Thus, these two LCs are sufficient to drive these behaviors. But are these LCs also necessary for looming evoked escape?*

We performed inactivation experiments and added the following text together with a new supplemental figure and legend (Figure 11—figure supplement 1):

New text in the Results:

“In view of these findings, we performed additional experiments to ask whether silencing of LC6 or LC16 reduces the amount of takeoff or backward walking in response to visual stimuli. […] Because multiple LC neuron types show jumping or backward walking responses to optogenetic stimulation (Figure 8) and non-LC cell types have also been reported to contribute to looming-evoked escape (de Vries and Clandinin, 2012), the simplest explanation of these results is that functional redundancy exists at the level of neurons mediating looming-evoked escape.”